# Multi-centre analysis of networks and genes modulated by hypothalamic stimulation in patients with aggressive behaviours

Flavia Venetucci Gouveia[1,2,3]*†, Jurgen Germann[4,5]†, Gavin JB Elias[4,5], Alexandre Boutet[4,6], Aaron Loh[4,5], Adriana Lucia Lopez Rios[7,8], Cristina Torres Diaz[9], William Omar Contreras Lopez[10,11], Raquel Chacon Ruiz Martinez[3,12], Erich Talamoni Fonoff[13], Juan Carlos Benedetti-Isaac[14], Peter Giacobbe[2,15,16], Pablo M Arango Pava[17], Han Yan[5,18], George M Ibrahim[5,18,19,20], Nir Lipsman[2,5,15], Andres Lozano[4,5], Clement Hamani[2,5,15]*

[1]Neuroscience and Mental Health, Hospital for Sick Children Research Institute, Toronto, Canada; [2]Sunnybrook Research Institute, Toronto, Canada; [3]Division of Neuroscience, Sírio-Libanês Hospital, São Paulo, Brazil; [4]Division of Neurosurgery, Department of Surgery, University Health Network, Toronto, Canada; [5]Division of Neurosurgery, Department of Surgery, University of Toronto, Toronto, Canada; [6]Joint Department of Medical Imaging, University of Toronto, Toronto, Canada; [7]Department of Functional and Stereotactic Neurosurgery, University Hospital San Vicente Fundación, São Paulo, Brazil; [8]Department of Functional and Stereotactic Neurosurgery, San Vicente Fundación, Barranquilla, Colombia; [9]Department of Neurosurgery, University Hospital La Princesa, Madrid, Spain; [10]Nemod Research Group, Universidad Autónoma de Bucaramanga, Bucaramanga, Colombia; [11]Division of Functional Neurosurgery, Department of Neurosurgery, FOSCAL Clinic, Bucaramanga, Colombia; [12]LIM 23, Institute of Psychiatry, School of Medicine, University of São Paulo, São Paulo, Brazil; [13]Department of Neurology, Integrated Clinic of Neuroscience, School of Medicine, University of São Paulo, São Paulo, Brazil; [14]Stereotactic and Functional Neurosurgery Division of the International Misericordia Clinic, Barranquilla, Canada; [15]Harquail Centre for Neuromodulation, Sunnybrook Health Sciences Centre, Toronto, Canada; [16]Department of Psychiatry, University of Toronto, Toronto, Canada; [17]Servicio de Neuocirugia Funcional y Esterotaxia, Clinica Comuneros Bucaramanga, Clinica Desa y Clinica Dime Neurocardiovascular de Cali; Clinica Nueva del Lago, Bucaramanga, Colombia; [18]Division of Neurosurgery, The Hospital for Sick Children, Toronto, Canada; [19]Institute of Biomedical Engineering, University of Toronto, Toronto, Canada; [20]Institute of Medical Science, University of Toronto, Toronto, Canada

*For correspondence:
fvenetucci@gmail.com (FVG);
clement.hamani@sunnybrook.ca (CH)

†These authors contributed equally to this work

Competing interest: The authors declare that no competing interests exist.

**Abstract** Deep brain stimulation targeting the posterior hypothalamus (pHyp-DBS) is being investigated as a treatment for refractory aggressive behavior, but its mechanisms of action remain elusive. We conducted an integrated imaging analysis of a large multi-centre dataset, incorporating volume of activated tissue modeling, probabilistic mapping, normative connectomics, and atlas-derived transcriptomics. Ninety-one percent of the patients responded positively to treatment, with

a more striking improvement recorded in the pediatric population. Probabilistic mapping revealed an optimized surgical target within the posterior-inferior-lateral region of the posterior hypothalamic area. Normative connectomic analyses identified fiber tracts and functionally connected with brain areas associated with sensorimotor function, emotional regulation, and monoamine production. Functional connectivity between the target, periaqueductal gray and key limbic areas – together with patient age – were highly predictive of treatment outcome. Transcriptomic analysis showed that genes involved in mechanisms of aggressive behavior, neuronal communication, plasticity and neuroinflammation might underlie this functional network.

## Editor's evaluation

This study presents useful structural and functional connectivity profiles of patients receiving deep brain stimulation in the posterior hypothalamus for severe and refractory aggressive behavior. The inclusion of data from multiple centers is compelling. This study will be important for a broad readership including basic and clinical neuroscientists.

## Introduction

Aggressive behaviors are highly prevalent among psychiatric patients, presenting a major obstacle to patient care. In addition to suffering, these symptoms constitute a leading cause for institutionalization *Gouveia et al., 2021a*; *Brentani et al., 2013*; *Gouveia et al., 2019*. Standard treatments for aggressive behaviors involve behavioral and pharmacological therapies that mainly act on the dopaminergic and serotonergic systems (e.g. serotonin reuptake inhibitors, antipsychotics) *Gouveia et al., 2021a*; *Brentani et al., 2013*; *Gouveia et al., 2019*. Despite their efficacy, a substantial proportion (30%) of patients fail to respond and are considered to be treatment refractory *Adler et al., 2015*; *Gouveia et al., 2020*; *Gouveia et al., 2021b*. For these patients, neuromodulation therapies, such as deep brain stimulation (DBS), have been investigated *Gouveia et al., 2019*; *Gouveia et al., 2021c*; *Contreras Lopez et al., 2021*; *Benedetti-Isaac et al., 2015*; *Torres et al., 2020*; *Micieli et al., 2017*; *López Ríos et al., 2023*. DBS is a neurosurgical therapy in which implanted electrodes are used to adjustably deliver electrical current to specific brain targets *Jakobs et al., 2019*. It is an established therapy for Parkinson's Disease, dystonia, essential tremor, epilepsy *Salanova et al., 2021* and obsessive-compulsive disorder *Hamani et al., 2014*; *Li et al., 2020*, showing promising results for the treatment of several other neuropsychiatric disorders, including Alzheimer's disease *Germann et al., 2021a*; *Lozano et al., 2016*, depression, anorexia nervosa, addiction, and posttraumatic stress disorder *Elias et al., 2021*; *Lozano et al., 2008*; *De Vloo et al., 2021*; *Elias et al., 2022*; *Hamani et al., 2022*; *Davidson et al., 2022*.

To date, DBS for refractory aggressive behavior has primarily targeted the posterior hypothalamic region *Gouveia et al., 2019*; *Gouveia et al., 2021c*; *Contreras Lopez et al., 2021*; *Benedetti-Isaac et al., 2015*; *Torres et al., 2020*; *Micieli et al., 2017*; *López Ríos et al., 2023*. The hypothalamus is a diencephalic structure with well-established roles in the control of homeostasis and motivated behaviors and is a key area in a broader neurocircuitry regulating aggressive behavior that also involves the orbitofrontal cortex, hippocampus, amygdala and periaqueductal gray *Dudás, 2013*; *Blair, 2016*; *Miczek et al., 2007*. Along the anterior-posterior axis, the hypothalamus can be divided into three regions (i.e. supraoptic or anterior, tuber cinereum or medial and supramamillary or posterior) with distinct cell types, projections and functions *Dudás, 2013*; *Beattie et al., 1930*. While the anterior region is mainly involved in thermoregulation and the control of circadian rhythms, the medial region regulates feeding and sexual behavior and plays a critical role in several endocrine and autonomic processes *Dudás, 2013*; *Hoff, 1950*; *Flament-Durand, 1980*. The posterior hypothalamus (pHyp) is an ergotropic area involved in the generation of sympathetic responses *Hess, 1945*. It provides robust projections to the midbrain and reticular formation via the hypothalamotegmental tract, thus being critical in the regulation of wakefulness and stress responses *Beattie et al., 1930*; *Lechan and Toni, 2016*; *Saper and Lowell, 2014*. Indeed, lesions (e.g. hamartomas and gliomas) located at the pHyp have been reported to cause apathetic and somnolent symptoms, while the selective neurosurgical ablation of a small portion of the pHyp has been successfully used to treat patients with severe

treatment-resistant aggressive behavior (for a detailed review, see *Gouveia et al., 2019*; *Sano et al., 1966*; *Sano and Mayanagi, 1988*).

More recently, DBS of the pHyp (pHyp-DBS) has also been shown to reduce aggressive symptoms in humans, with variable outcomes *Gouveia et al., 2021c*; *Contreras Lopez et al., 2021*; *Benedetti-Isaac et al., 2015*; *Torres et al., 2020*; *Franzini et al., 2010*; *Torres et al., 2013*; *López Ríos et al., 2023*. At present, only 21patients diagnosed with autism spectrum disorder (ASD), intellectual disability, obsessive-compulsive disorder, epilepsy and schizophrenia with ages ranging from 10 to 51 years have been reported, mostly as individual cases or small case series. In these patients, long-term improvement (up to several years of follow-up) in aggressive behaviors was in the order of 38–100% compared to baseline *Gouveia et al., 2021c*; *Contreras Lopez et al., 2021*; *Benedetti-Isaac et al., 2015*; *Torres et al., 2020*; *Micieli et al., 2017*; *López Ríos et al., 2023*. Although case studies and small series provide valuable insight into the safety and therapeutic impact of pHyp-DBS in individual patients, they do not allow the characterization of clinical phenotypes, optimal stimulation targets or brain networks underlying treatment.

In this work, we gathered data from the largest international multi-centre dataset of patients treated to date with pHyp-DBS for aggressive behaviors to retrospectively investigate possible neuro-biological mechanisms of action. Combining a well-documented electrode localization and volume of activated tissue (VAT) modeling pipeline (https://www.lead-dbs.org/) with probabilistic sweet-spot mapping *Elias et al., 2021*; *Dembek et al., 2017*, normative connectomics *Fox, 2018*; *Elias et al., 2020*; *Germann et al., 2021b*, and transcriptomics analysis *Gouveia et al., 2021a*; *Mroczek et al., 2021* (https://alleninstitute.org/), we delineated a potentially 'optimized' surgical target and identified the brain networks and underlying neurobiological processes that might underpin successful pHyp-DBS. Demographic data, pre-and post-operative magnetic resonance imaging (MRI) and computed tomography (CT) scans were obtained from each participant for DBS lead localization, followed by an estimation of the VAT and the determination of the most effective region of stimulation. The VATs were further processed for connectomic analyses to investigate the structural (i.e. fiber tracts) and functional (i.e. brain areas) maps associated with symptom improvement. Using demographics and individual functional connectivity, we tested a predictive model of improvement following pHyp-DBS. Finally, we investigated genes with a spatial pattern of distribution similar to the functional connectivity map to explore associated biological processes. A graphical summary of the methodology used in this study can be found in *Figure 1* and in *Figure 1—figure supplement 1*.

## Results
### Patients included
In this retrospective study, we aggregated a large dataset of 33 patients from 5 centers treated with pHyp-DBS to alleviate intractable aggressive behaviors, characterized by self-injurious and extreme aggressive behaviors towards their surroundings and others (12 females and 21 males, 24.48±10.28 years of age ranging from 10 to 52y, *Figure 2A–B*). The most frequent diagnoses were epilepsy (pediatric: 50%, adult: 62%) and ASD (pediatric: 34%, adult: 24%). Intellectual disability was observed in all cases. This was described as severe in 82% of patients (pediatric: 75%, adult: 85%) and moderate in 18% of patients (pediatric: 25%, adult: 15%). *Table 1* presents demographic data. Clinical trials and individual cases were evaluated by the corresponding local ethics committee, and informed consent was obtained *Gouveia et al., 2021c*; *Contreras Lopez et al., 2021*; *Benedetti-Isaac et al., 2015*; *Torres et al., 2020*; *Micieli et al., 2017*; *Torres et al., 2013*; *López Ríos et al., 2023*; . Surgical treatment was approved on a humanitarian basis, given the chronicity and severity of symptoms and the lack of response to conservative treatment.

In all centers, patients were evaluated by a multidisciplinary healthcare team that reviewed all clinical and medication history. Patients were only considered for surgery when a consensus was reached that they indeed presented severe medication-resistant aggressive behavior (i.e. persistent severe symptomatology despite using multiple medications at well-established doses and duration). Whole-brain T1-weighted MRI was acquired preoperatively for surgical planning (1.5T MRI used in 14 cases at 3 centers, and 3T MRI used in 19 cases at 2 centers). Postoperative brain MRI and/or CT were obtained for confirmation of electrode localization. Aggressive behavior was assessed using standard questionnaires conducted by a neuropsychologist and answered by the parents/caregivers

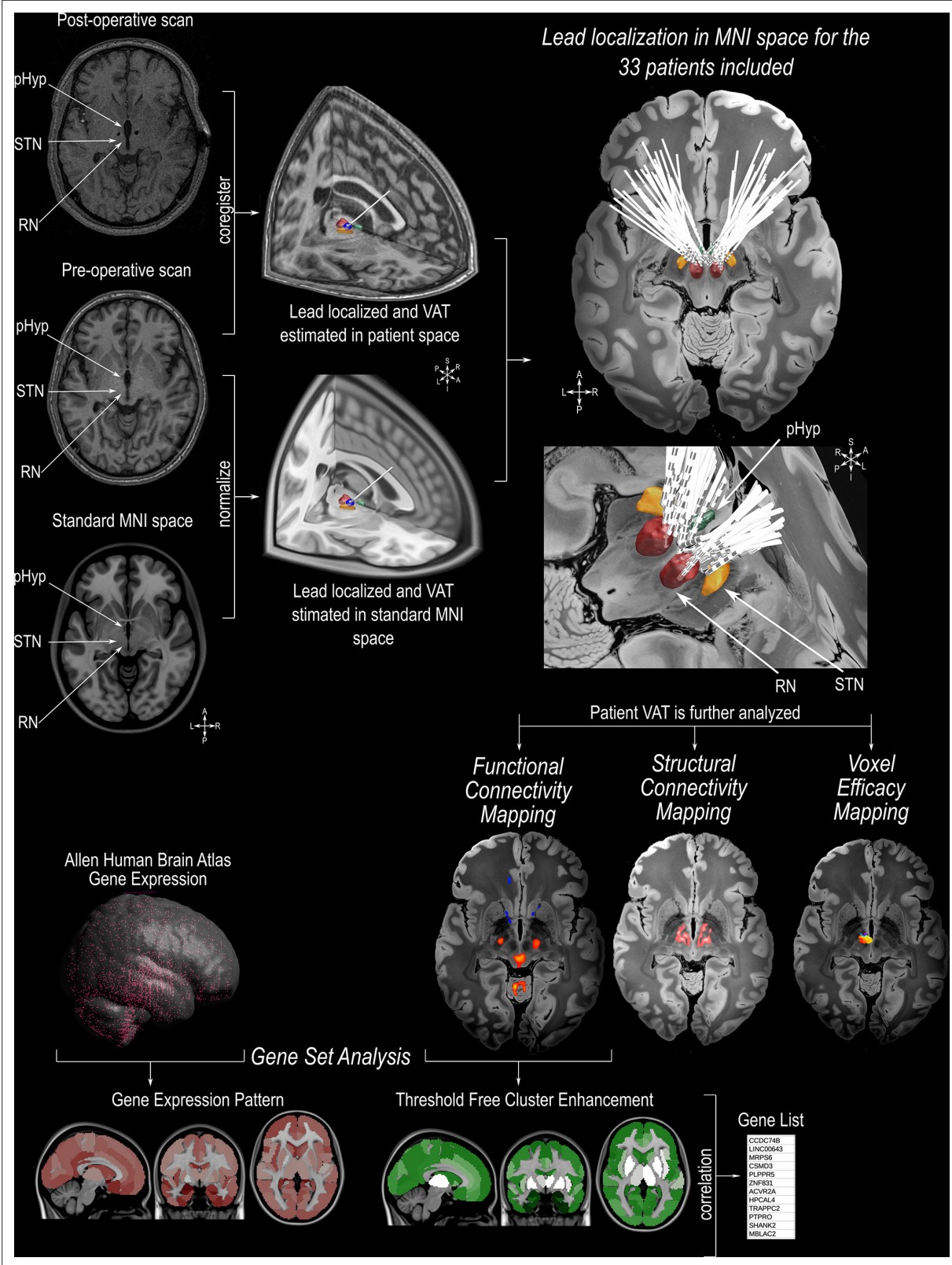

**Figure 1.** Illustration of the methodologies applied in this study. Preoperative MRI scans were co-registered with the postoperative MRI/CT scan, followed by normalization to standard MNI152 space (https://www.bic.mni.mcgill.ca/ServicesAtlases/ICBM152NLin2009). Individual DBS leads were manually localized in the posterior hypothalamic area (pHyp) in the patient space and normalized to MNI152 space. The estimation of the volume of activated tissue (VAT) was calculated based on individual stimulation parameters using Lead-DBS (https://www.lead-dbs.org/ See *Table 1* for individual

*Figure 1 continued on next page*

*Figure 1 continued*

stimulation parameters). The patients' VATs were further investigated for the analysis of the Voxel Efficacy Map (determination of the optimal stimulation site), Imaging Connectomics using Structural Connectivity Map (determining the streamlines involved in symptom improvement) and Functional Connectivity Map (determining the functionally connected areas involved in symptom improvement). For imaging Transcriptomics, we applied a Threshold Free Cluster Enhancement (TFCE) to the functional connectivity map. Functionally connected areas were averaged into the Harvard-Oxford Atlas (http://www.cma.mgh.harvard.edu/). Based on the human gene expression data from the Allen Human Brain Atlas (https://alleninstitute.org/), genes with a spatial pattern distribution similar to the TFCE map were selected for further gene ontology analysis. 3D reconstruction of the DBS leads on a 100micron resolution, 7.0 Tesla FLASH brain (https://openneuro.org/datasets/ds002179/versions/1.1.0) in MNI152 space; the pHyp label was derived from a previously published high-resolution MRI atlas of the human hypothalamic region (https://zenodo.org/record/3903588#.YHiE7pNKiF0).

The online version of this article includes the following figure supplement(s) for figure 1:

**Figure supplement 1.** Method of generating functional connectivity maps.

in all centers (i.e. Overt Aggressive Behaviour [OAS] used in 3 centers; Modified Overt Aggressive Behaviour [MOAS] used in 1 center; Inventory for Client and Agency Planning [ICAP] used in 1 center). Treatment response is reported as the percentage of improvement at the last follow-up relative to baseline (preoperative). Patients presenting more than 30% improvement were considered to be treatment responders.

pHyp-DBS was bilaterally implanted in 30 patients and unilaterally implanted in 3 using either Medtronic Activa (3387 leads in 19 cases and 3389 leads in 10 cases) or Boston Vercise DBS Systems (4

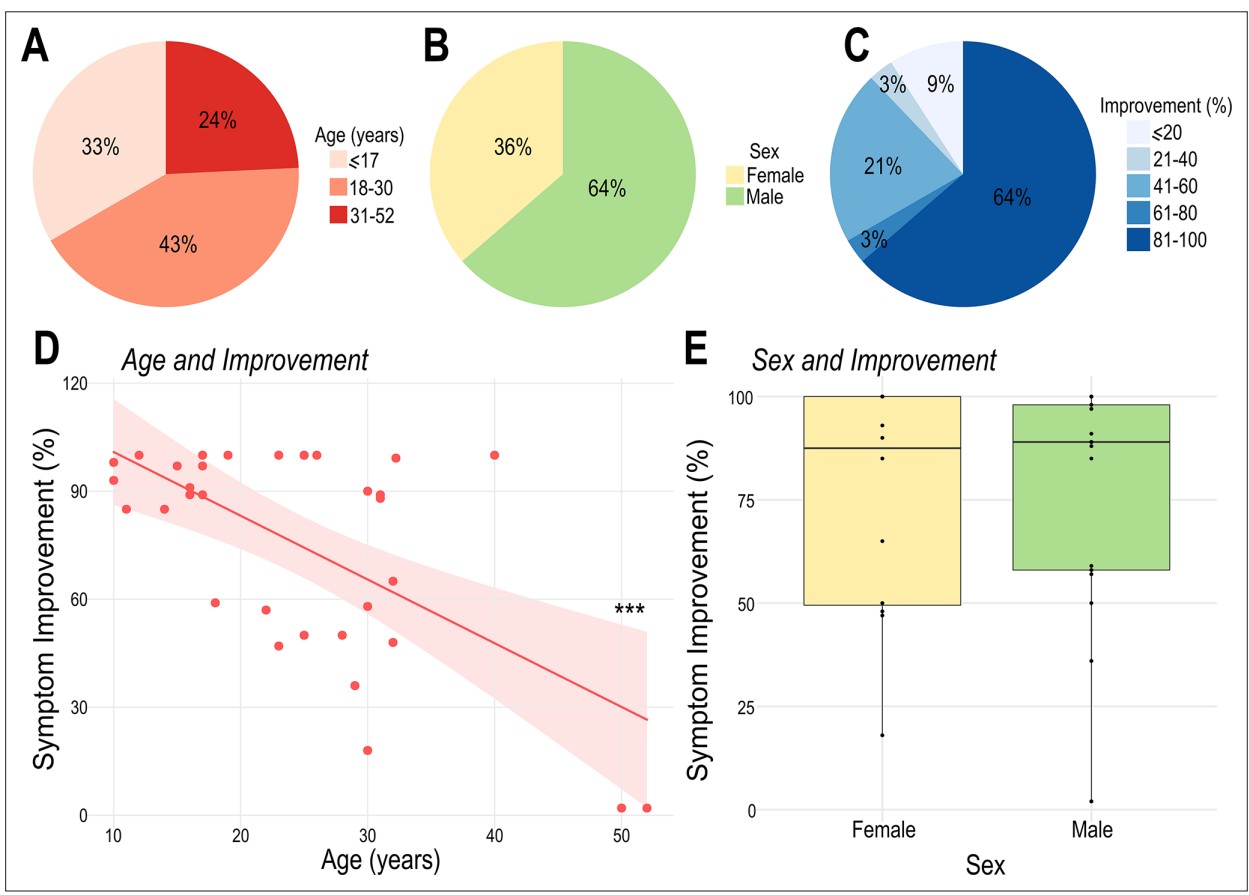

**Figure 2.** Patient demographics and treatment outcome. (**A**) Patients were divided in three main groups according to age: pediatric population (≤17years, 11 out of 33), young adults (18–30years, 14 out of 33) and older adults (31–52years, 8 out of 33). (**B**)Distribution of males (21 out of 33) and females (12 out of 33) in this study. (**C**)Patient distribution according to the percentage of symptomatic improvement (≤20: 3 out of 33; 21–40: 1 out of 33; 41–60: 7 out of 33; 61–80: 1 out of 33; 81–100: 21 out of 33). Note that the majority of individuals presented over 30% improvement following treatment (criteria for being considered a treatment responder), and a large proportion of patients presented an improvement greater than 80%. (**D**)Age at surgery was significantly negatively correlated with postoperative symptomatic improvement (R=−0.61; R2=0.38; *** p<0.001). (**E**)There was no significant difference in the percentage of symptomatic improvement between male and female patients.

**Table 1.** Demographics.

| Case | Sex | Age range | Improvement (%) | Laterality | DBS System | Stimulation Settings Right | Left |
|------|-----|-----------|-----------------|------------|------------|-------|------|
| 1 | M | 31–52 | 100 | Bilateral | Medtronic 3387 | 1.8V; 180Hz; 60msec | 1.8V; 180Hz; 60msec |
| 2 | M | ≤17 | 97 | Bilateral | Medtronic 3387 | 2.2V; 200Hz; 90msec | 2.2V; 200Hz; 90msec |
| 3 | M | ≤17 | 98 | bilateral | Medtronic 3387 | 2.5V; 180Hz; 90msec | 2.5V; 180Hz; 90msec |
| 4 | M | ≤17 | 89 | Bilateral | Medtronic 3387 | 5.0V; 210Hz; 90msec | 5.0V; 210Hz; 90msec |
| 5 | F | ≤17 | 93 | Bilateral | Medtronic 3387 | 2.0V; 200Hz; 90msec | 2.0V; 200Hz; 90msec |
| 6 | F | ≤17 | 85 | Bilateral | Medtronic 3387 | 4.0V; 180Hz; 90msec | 4.0V; 180Hz; 90msec |
| 7 | F | ≤17 | 100 | Bilateral | Medtronic 3387 | 3.5V; 180Hz; 90msec | 3.5V; 180Hz; 90msec |
| 8 | F | 31–52 | 100 | Bilateral | Medtronic 3387 | 5.0V; 200Hz; 100msec | 5.0V; 200Hz; 100msec |
| 9 | F | 18–30 | 100 | Bilateral | Medtronic 3387 | 2.3V; 200Hz; 120msec | 2.3V; 200Hz; 120msec |
| 10 | M | ≤17 | 85 | Bilateral | Medtronic 3387 | 2.0V; 130Hz; 60msec | 2.0V; 130Hz; 130msec |
| 11 | M | 18–30 | 100 | Bilateral | Medtronic 3387 | 3.0V; 180Hz; 90msec | 3.0V; 180Hz; 90msec |
| 12 | M | ≤17 | 89 | Bilateral | Medtronic 3387 | 5.5V; 185Hz; 130msec | 5.5V; 185Hz; 130msec |
| 13 | F | 18–30 | 47 | Bilateral | Medtronic 3387 | 7.0V; 250Hz; 120msec | 7.0V; 250Hz; 120msec |
| 14 | M | 18–30 | 100 | Bilateral | Medtronic 3387 | 5.0V; 210Hz; 130msec | 5.0V; 210Hz; 130msec |
| 15 | F | 18–30 | 90 | Bilateral | Medtronic 3387 | 3.5V; 180Hz; 90msec | 3.5V; 180Hz; 90msec |
| 16 | M | 18–30 | 100 | Bilateral | Medtronic 3387 | 5.0V; 200Hz; 100msec | 5.0V; 200Hz; 100msec |
| 17 | F | ≤17 | 100 | Bilateral | Medtronic 3387 | 3.5V; 180Hz; 90msec | 3.5V; 180Hz; 90msec |
| 18 | M | ≤17 | 97 | Bilateral | Medtronic 3387 | 4.5V; 180Hz; 90msec | 4.5V; 180Hz; 90msec |
| 19 | M | 31–52 | 88 | Bilateral | Medtronic 3389 | 3.0V; 180Hz; 90msec | 3.0V; 180Hz; 90msec |
| 20 | M | ≤17 | 91 | Bilateral | Medtronic 3389 | 3.2V; 180Hz; 90msec | 3.2V; 180Hz; 90msec |
| 21 | F | 18–30 | 18 | Bilateral | Medtronic 3389 | 3.8V; 180Hz; 90msec | 3.8V; 180Hz; 90msec |
| 22 | M | 31–52 | 89 | Bilateral | Medtronic 3389 | 3.5V; 180Hz; 90msec | 3.5V; 180Hz; 90msec |
| 23 | M | 31–52 | 2 | Bilateral | Medtronic 3389 | 4.5V; 150Hz; 247msec | 4.5V; 150Hz; 247msec |
| 24 | M | 18–30 | 57 | Unilateral | Medtronic 3389 | 0.3V; 150Hz; 450msec | Not applicable |

*Table 1 continued on next page*

*Table 1 continued*

| Case | Sex | Age range | Improvement (%) | Laterality | DBS System | Stimulation Settings | |
|------|-----|-----------|------------------|------------|------------|-------|------|
| | | | | | | Right | Left |
| 25 | M | 31–52 | 2 | Unilateral | Medtronic 3389 | 0.9V; 150Hz; 450msec | Not applicable |
| 26 | F | 31–52 | 65 | Bilateral | Medtronic 3389 | 0.1V; 60Hz; 180msec | 0.1V; 60Hz; 300msec |
| 27 | M | 18–30 | 59 | Bilateral | Medtronic 3389 | 0.7V; 150Hz; 330msec | 0.5V; 150Hz; 450msec |
| 28 | F | 31–52 | 48 | Unilateral | Medtronic 3389 | 0.1V; 150Hz; 450msec | Not applicable |
| 29 | M | 18–30 | 100 | Bilateral | Medtronic 3387 | 2.0V; 180Hz; 120msec | 2.0V; 180Hz; 120msec |
| 30 | M | 18–30 | 50 | Bilateral | Boston, Vercise | 1.0mA; 185Hz; 90msec | 1.0mA; 185Hz; 90msec |
| 31 | F | 18–30 | 50 | Bilateral | Boston, Vercise | 1.2mA; 113Hz; 120msec | 1.2mA; 113Hz; 120msec |
| 32 | M | 18–30 | 36 | Bilateral | Boston, Vercise | 1.0mA; 170Hz; 70msec | 1.0mA; 170Hz; 70msec |
| 33 | M | 18–30 | 58 | Bilateral | Boston, Vercise | 3mA; 185Hz; 60msec | 3mA; 185Hz; 60msec |

To preserve patients' anonymization, the diagnoses observed in this group are presented as the following list, from more to less frequent. Epilepsy, autism spectrum disorder, tuberous sclerosis, congenital rubella, intermittent explosive disorder, agenesia of the corpus callosum, schizophrenia, obsessive-compulsive disorder, West syndrome, Landau-Kleffner syndrome, Cri-du-chat syndrome, Lennox-Gastaut syndrome, Sotos syndrome, meningoencephalitis, perinatal hypoxia, periventricular leucomalacia, microcephaly, arteriovenous malformation.

cases). Average stimulation parameters for the right hemisphere were 2.45V±1.76V (range 0.3–4.5V), 166.25Hz±18.87Hz (range 150–185Hz) and 219.25µsec ± 172.46µsec (range 60 to 450µsec). For the left hemisphere, average stimulation parameters were 3.17V±1.26V (range 2–4.5V), 171.67Hz±18.93Hz (range 150–185Hz) and 142.33µsec ± 95.48µsec (range 60 to 257µsec). Individual stimulation parameters are shown in *Table 1*.

After treatment, 91% (30 out of 33) of the patients were considered to be responders (>30% decrease in validated scores compared to baseline) *Adler et al., 2015*; *Gouveia et al., 2020*; *Gouveia et al., 2021b*; *Benedetti-Isaac et al., 2015*. The average percentage of improvement was 75.25 ± 29.59% (*Figure 2C*). Younger patients were found to have a more pronounced benefit, with the pediatric population (patients ≤ 17 years of age) exhibiting greater symptom improvement compared to the adult population (93% vs 66%, *Figure 2D*). No differences were observed between males (75.57% ± 31.13%) and females (74.68% ± 28.00%; *Figure 2E*).

## Probabilistic sweet-spot mapping

To provide insight into the relationship between stimulation location and response to pHyp-DBS treatment, probabilistic maps of efficacious stimulation were generated using previously described methods *Elias et al., 2021*; *Dembek et al., 2017*. Briefly, preoperative MRI scans were co-registered to individual postoperative MRI/CT scans, normalized to standard MNI152 space for estimation of the VAT. Left-sided VATs were flipped at the sagittal plane, weighted by the corresponding percentage of improvement and the mean improvement of overlapping VATs was calculated at each voxel. These average maps were then thresholded for voxel-wise significance using a Wilcoxon signed rank test (p<0.05). Additionally, to exclude outlier voxels, only those detected in>10% of all maps were included. Finally, we performed nonparametric permutation testing, randomly assigning each clinical score to a VAT, as previously described *Dembek et al., 2017*; *Eisenstein et al., 2014*; *Dembek et al., 2019*.

This analysis revealed greater symptom alleviation related to the stimulation of a more posterior-inferior-lateral region of the posterior hypothalamic area (*Figure 3*). This area encompassed 684

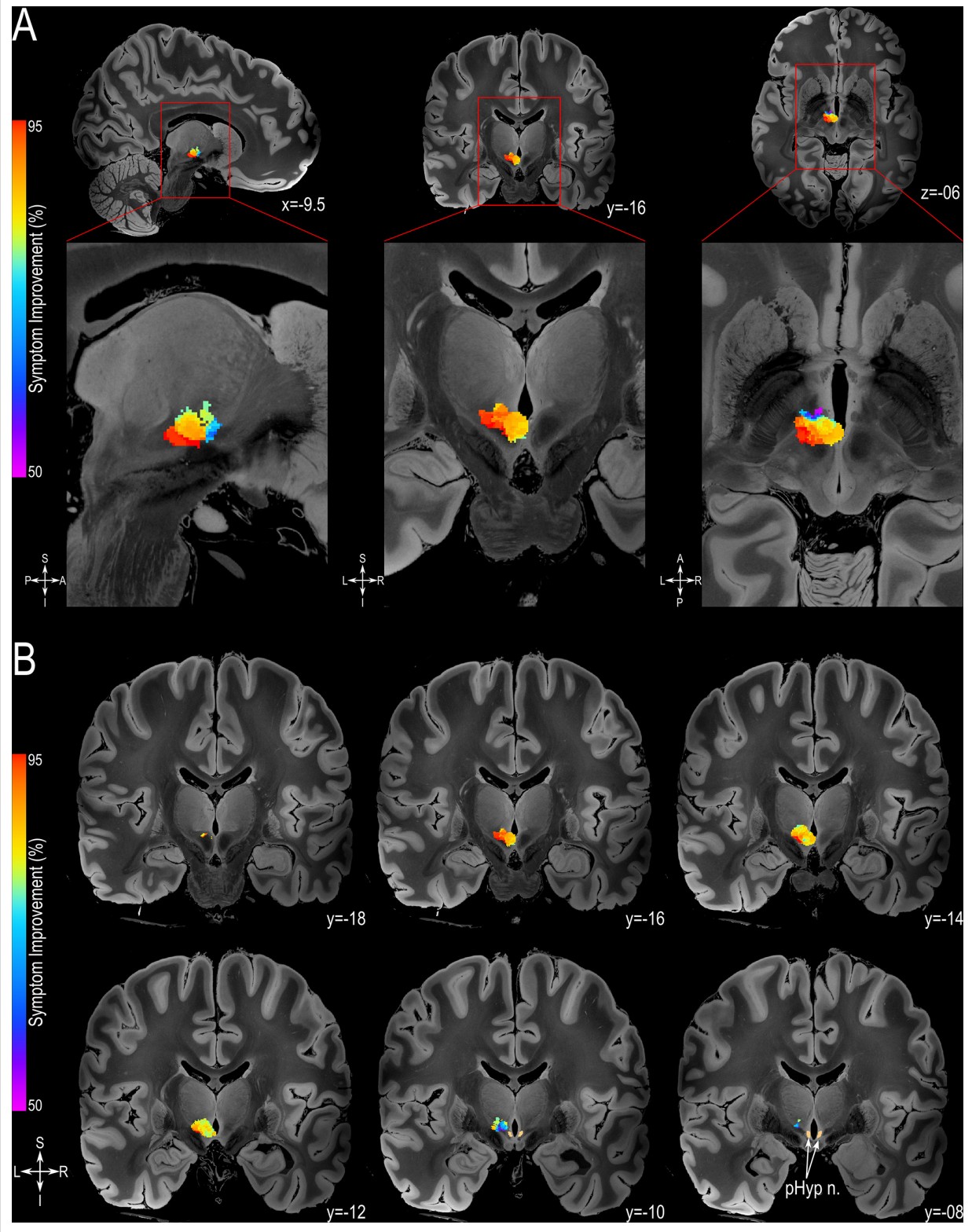

**Figure 3.** Probabilistic Sweet-spot Mapping. (**A**) The area of stimulation associated with greater symptomatic improvement (red) was located in the more posterior-inferior-lateral region of the posterior hypothalamic area (from left to right: sagittal, coronal and axial views). (**B**)The extent of the volumes of activated tissue (VATs) responsible for eliciting at least 50% improvement is shown in successive coronal MRI slices. All results are illustrated on slices of a 100micron resolution, 7.0 Tesla FLASH brain (https://openneuro.org/datasets/ds002179/versions/1.1.0) in MNI152 space (https://www.bic.

*Figure 3 continued on next page*

Figure 3 continued

mni.mcgill.ca/ServicesAtlases/ICBM152NLin2009). The posterior hypothalamic nucleus (pHyp n.) label (shown in beige) was derived from a previously published high-resolution MRI atlas of the human hypothalamic region (https://zenodo.org/record/3903588#.YHiE7pNKiF0).

The online version of this article includes the following figure supplement(s) for figure 3:

**Figure supplement 1.** Localization of the probabilistic sweet spot mapping associated with at least 50% improvement in symptoms, in the posterior-inferior-lateral region of the posterior hypothalamic area, and in relation to the red nucleus and the subthalamic nucleus.

**Figure supplement 2.** Comparison between two probabilistic sweet-spot maps performed considering amplitude (top panel, original analysis) and amplitude plus pulse width (bottom panel, additional analysis).

voxels associated with a behavioral improvement greater than 90%. The centroid of this highly efficacious area can be found at MNI152 space at the coordinates x=7.5, y=-15, z=−6.5 (https://www.bic.mni.mcgill.ca/ServicesAtlases/ICBM152NLin2009) and in the Talairach-Tournoux space (http://www.talairach.org/) at coordinates x=6.5, y=-16, z=−1.5. The permutation test showed that this pattern indeed reflects the specific relationship of individual VATs with the individual outcome ($p_{permute}$ <0.01).

## Normative connectomics analyses - structural and functional connectivity mapping

To investigate white matter tracts and brain networks associated with symptom improvement, normative structural and functional connectivity mapping were performed *Fox, 2018*; *Elias et al., 2020*; *Germann et al., 2021b*. Structural connectivity mapping employed diffusion MRI-based tractography data sourced from Human Connectome Project subjects to identify streamlines that intersected individual VATs. Functional connectivity mapping used Genomics Superstruct Project-derived resting-state functional MRI information to generate voxel-wise correlation maps that reflected each VTA's brain-wide functional connectivity. Subsequent statistical analyses were conducted to determine which streamlines and functional connectivity patterns were associated with positive treatment outcomes.

A variety of streamline bundles were identified by this analysis as being clinically relevant, such that VATs touching said streamlines corresponded to better outcomes than VATs that did not. These streamlines bundles may be divided into three main categories of function: I. Somatosensation (Medial Lemniscus and Spinothalamic Tract); II. Regulation of emotions (Amygdalofugal Pathway, Anterior Limb of the Internal Capsule, Medial Forebrain Bundle *Nieuwenhuys et al., 2007*; *Nieuwenhuys et al., 1982*); III. Motor Connections (Superior Cerebellar Peduncle, Rubrospinal tract, Frontopontine tract, Central Tegmental Tract, Medial-longitudinal Fasciculus, Motor Projections; *Figure 4A–B*, *Figure 4—figure supplement 1*). Interestingly, when overlapping the voxel efficacy and the structural connectivity maps (*Figure 4C*), we observed that voxels associated with higher efficacy (in red) were more closely related to these fiber tracts.

The functional connectivity analysis showed that the extent of VAT connectedness to several areas was significantly associated with clinical benefits. These areas are related to the production of monoamines (i.e. dorsal and medial raphe nuclei [serotonin]; substantia nigra [dopamine]; *Figure 5*) and are known to be components of the neurocircuitry of aggressive behavior (e.g. amygdala; nucleus accumbens, rostral anterior cingulate cortex; bed nucleus of the stria terminalis; hypothalamus; dorsal anterior cingulate cortex, insula, periaqueductal grey; *Figure 5*) *Gouveia et al., 2021a*; *Gouveia et al., 2019*; *Gouveia et al., 2020*; *Gouveia et al., 2021b*. Functional connectivity mapping after Threshold-Free Cluster Enhancement (TFCE) analysis FDR corrected at q<0.0001 is presented in *Figure 5—figure supplement 1*.

## Estimation of clinical outcome

To investigate whether individual functional connectivity to particular hubs within the neurocircuitry of aggressive behavior could be used to estimate symptom improvement following pHyp-DBS, additive linear models were created. For this, we extracted individual connectivity values from the peak within each brain area where functional connectivity with the VATs was found to be significantly related to outcomes at the group level (as described in the Normative Connectomics Analyses, *Figure 6A*). The best-performing parsimonious model, which incorporated patient age as well as individual VAT functional connectivity, revealed the Periaqueductal Grey Matter (PAG) to be the only structure that, together with age, significantly predicted more than half of the variance in individual symptom

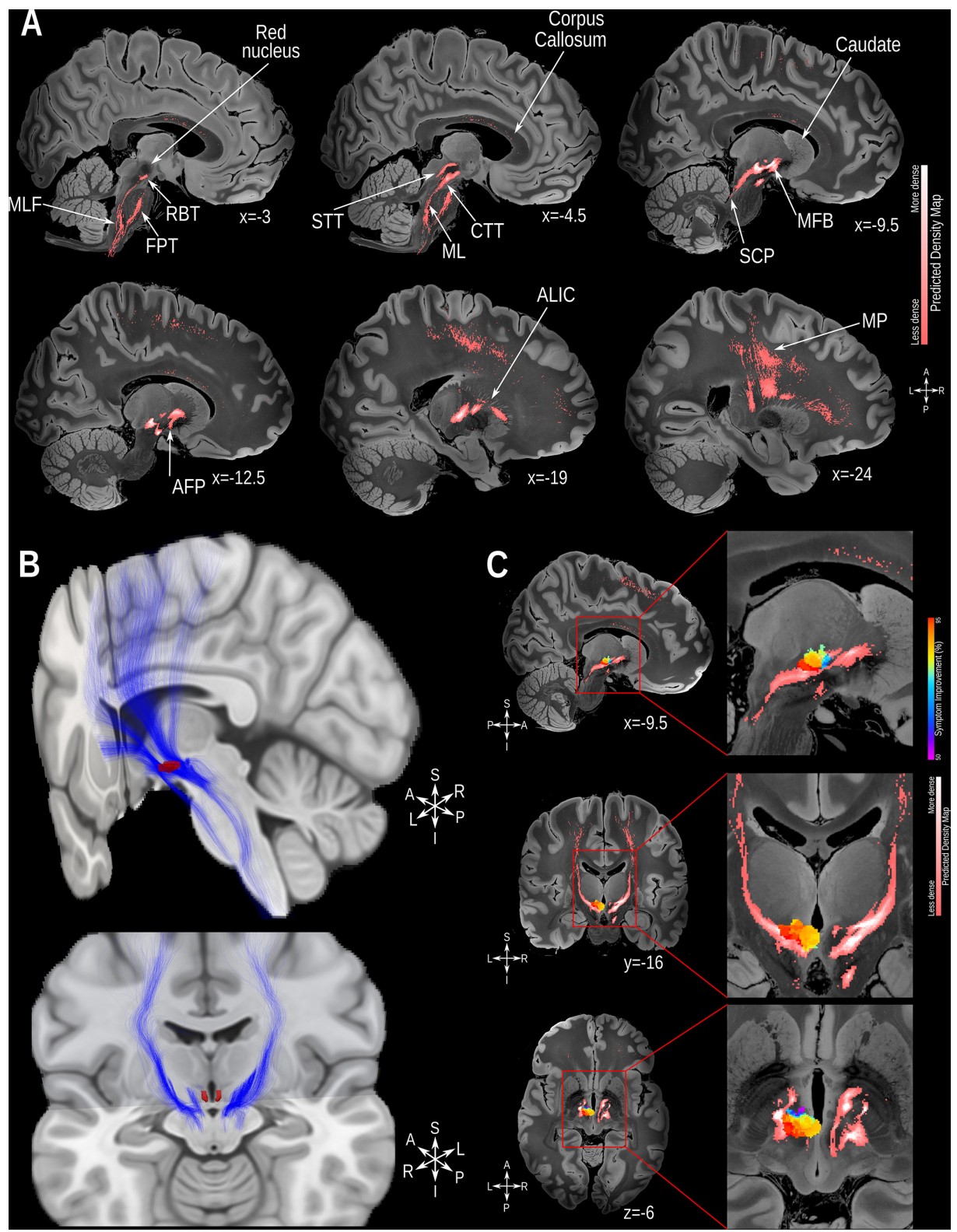

**Figure 4.** Structural connectivity mapping. (**A**) Magnetic resonance imaging (MRI) in the sagittal plane showing the fiber density of streamlines connected to the volumes of activated tissue (VATs) associated with significantly greater symptomatic improvement. (**B**)3D reconstruction of the streamlines associated with significantly greater improvement illustrated on the MNI152 brain (https://www.bic.mni.mcgill.ca/ServicesAtlases/ICBM152NLin2009); the posterior hypothalamic nucleus label (in red) was derived from a previously published high-resolution MRI atlas of the human

*Figure 4 continued on next page*

*Figure 4 continued*

hypothalamic region (https://zenodo.org/record/3903588#.YHiE7pNKiF0). (**C**)MRI showing the relation between VATs responsible for eliciting at least 50% improvement and the fiber density map (from top to bottom: sagittal, coronal and axial views). The results presented in A and C are illustrated on a 100micron resolution, 7.0 Tesla FLASH brain (https://openneuro.org/datasets/ds002179/versions/1.1.0) in MNI152 space. Abbreviations: AFP: Amygdalofugal Pathway; ALIC: Anterior Limb of the Internal Capsule; CTT: Central-Tegmental Tract; FPT: Frontopontine Tract; MFB: Medial Forebrain Bundle; ML: Medial Lemniscus; MLF: Medial-Longitudinal Fasciculus; MP: Motor Projections; RBT: Rubrospinal Tract; SCP: Superior Cerebellar Peduncle; STT: Spino-Thalamic Tract.

The online version of this article includes the following figure supplement(s) for figure 4:

**Figure supplement 1.** Structural connectivity mapping.

improvement ($R$=0.72, $R^2$=0.52, p=1.86e-05, *Figure 6B*, *Table 2*), retaining significance during leave-one-out cross-validation (LOOCV) ($R$=0.65, $R^2$=0.42, p=4.406e-5; *Figure 6C*). Additionally, we investigated if functional connectivity of the VATs with any 2 brain areas, in addition to patient age, would improve the estimation of clinical outcome (*Supplementary file 1*). Indeed, connectivity with PAG and limbic structures, namely the amygdala ($R$=0.75, $R^2$=0.57, p=4.20e-07), anterior cingulate cortex (rostral: $R$=0.76, $R^2$=0.58, p=2.75e-07; dorsal: $R$=0.75, $R^2$=0.57, p=3.96e-07), bed nucleus of the stria terminallis ($R$=0.76, $R^2$=0.58, p=2.75e07), left nucleus accumbens ($R$=0.76, $R^2$=0.58, p=8.51e-07), right orbitofrontal cortex ($R$=0.76, $R^2$=0.58, p=2.75e-07) and right fusiform gyrus ($R$=0.75, $R^2$=0.56, p=4.67e-07) was superior in predicting outcome.

## Imaging transcriptomics – Gene set analysis

Finally, to investigate neural phenotypes and possible neurobiological mechanisms of treatments, we performed imaging transcriptomics analysis using the abagen toolbox (https://abagen.readthe-docs.io/en/stable/index.html) and the human gene expression data from the Allen Human Brain Atlas (https://alleninstitute.org/). We investigated genes with a spatial distribution of expression that resembled the pattern of brain regions with clinically relevant (following TFCE correction) functional connectivity to the stimulation locus ($q_{FDRcor}$ <0.0001). This type of analysis shows genes whose spatial pattern correlates with brain changes that are also found in post-mortem and large-scale GWAS studies of specific patient populations *Arnatkeviciute et al., 2022*. This process resulted in an extensive list of candidate genes that were Bonferroni corrected (p<0.005) before being investigated using the EnRichr Gene Ontology tool (https://maayanlab.cloud/Enrichr/) and a cell-specific aggregate gene set *Seidlitz et al., 2020* to identify associated biological processes and cell types (*Figure 7*). We observed genes associated with clinically relevant brain areas considered to be at the core of the neurocircuitry of aggressive behavior (i.e. Hypothalamus, Amygdala, Prefrontal Cortex and Cingulate Cortex) *Gouveia et al., 2021a*; *Gouveia et al., 2019*; *Gouveia et al., 2020*; *Gouveia et al., 2021b*; *Blair, 2016*, genes linked to oxytocin (a peptide hormone critically implicated in social behaviors) *Ne'eman et al., 2016*; *Johansson et al., 2012*; *Neumann, 2008*; *Malik et al., 2012*; *Amorim et al., 2018* and to cell compartments related to neuronal communication and plasticity (e.g. neuron projection, synapse; axon guidance, long-term potentiation). Additionally, investigating the prevalence of cell types associated with the set of genes identified in our analysis, we found a significant overrepresentation of genes associated with oligodendrocytes and an underrepresentation of genes associated with astrocytes and microglia when compared to the expected distribution of genes per cell type (*Figure 7*).

## Discussion

In this work, we performed an integrated imaging analysis of a large, international multi-center dataset of patients treated with pHyp-DBS for severe refractory aggressive behavior. Overall, the response rate for pHyp-DBS was high, with 91% (30 out of 33) of patients qualifying as treatment responders. In addition, 64% of patients presented symptom improvement greater than 81% (*Figure 2*), and the average improvement across the whole pHyp-DBS cohort was 75%. These results are in line with previous work showing that symptom improvement following hypothalamic neurosurgery for the treatment of aggressive behavior is in the order of 90% *Gouveia et al., 2019*. In our series, four patients were implanted with directional DBS electrodes. Outcomes in patients implanted with these systems or conventional electrodes were found to be comparable *Contreras Lopez et al., 2021*.

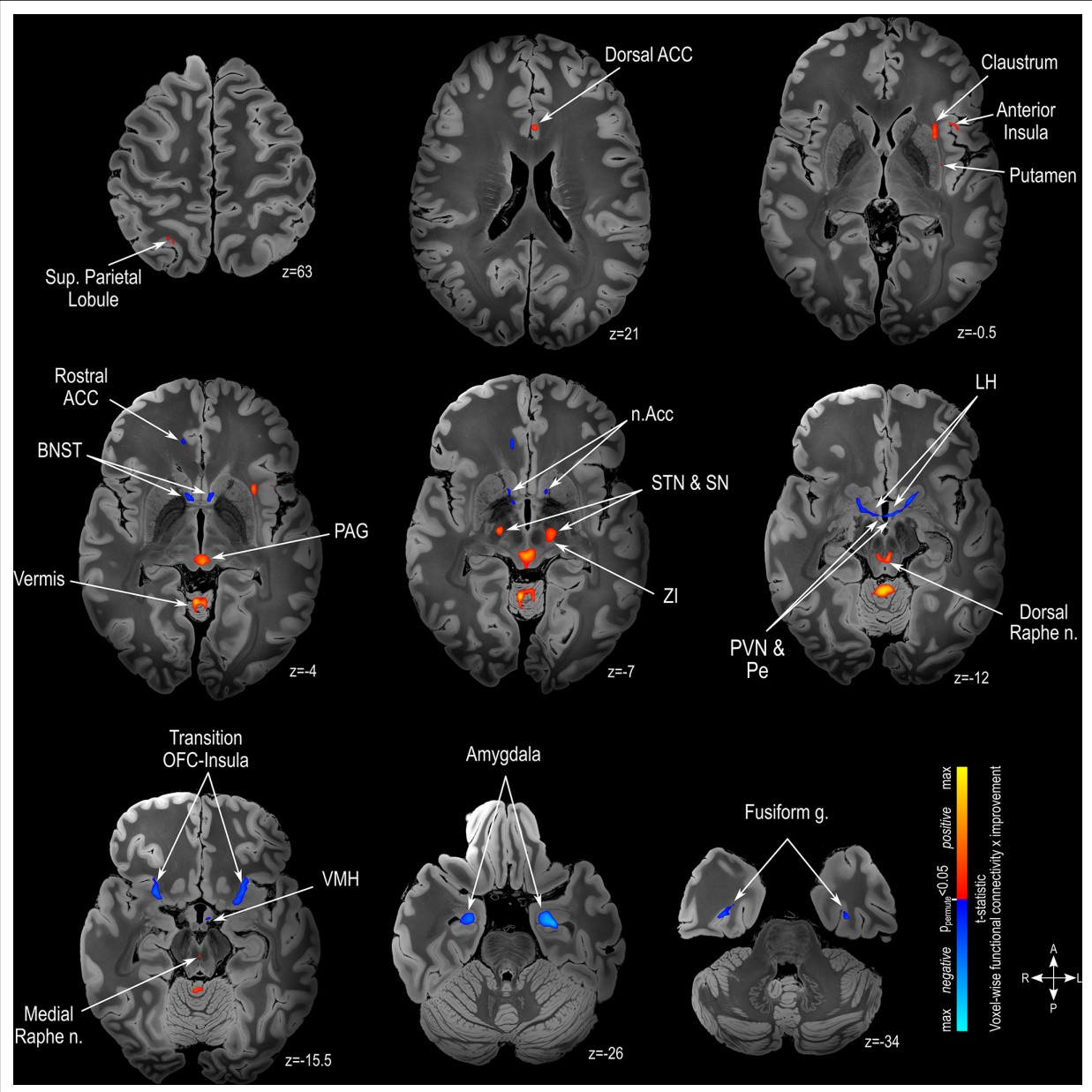

**Figure 5.** Functional connectivity mapping. Magnetic resonance imaging (MRI) in the axial plane showing areas found to have a positive (warm colors) or a negative (cold colors) correlation between clinical outcome and functional connectivity. Results are illustrated on a 100micron resolution, 7.0 Tesla FLASH brain in MNI152 space (https://openneuro.org/datasets/ds002179/versions/1.1.0). Abbreviations: ACC: Anterior Cingulate Cortex; BNST: Bed Nucleus of Stria Teminalis; LH: Lateral Hypothalamus; n.Acc: Nucleus Accumbens; OFC: Orbitofrontal Cortex; PAG: Periaqueductal Grey matter; Pe: Periventricular Hypothalamus; PVN: Paraventricular Hypothalamus; SN: Substantia Nigra; STN: Subthalamic Nucleus; VMH: Ventromedial Hypothalamus; ZI: Zona Incerta.

The online version of this article includes the following figure supplement(s) for figure 5:

**Figure supplement 1.** Threshold-free cluster enhancement functional connectivity mapping.

While analysis derived from retrospective clinical data is more prone to potential bias, this study included a collection of previously published prospective trials, where each center standardized the timeframe for data collection and the instruments used to evaluate symptom improvement.

Although we did not observe a difference in the main diagnosis between the pediatric and adult populations, the former presented the highest percentage of symptom improvement following pHyp-DBS treatment. Considering that the brain only fully achieves maturity in early adulthood

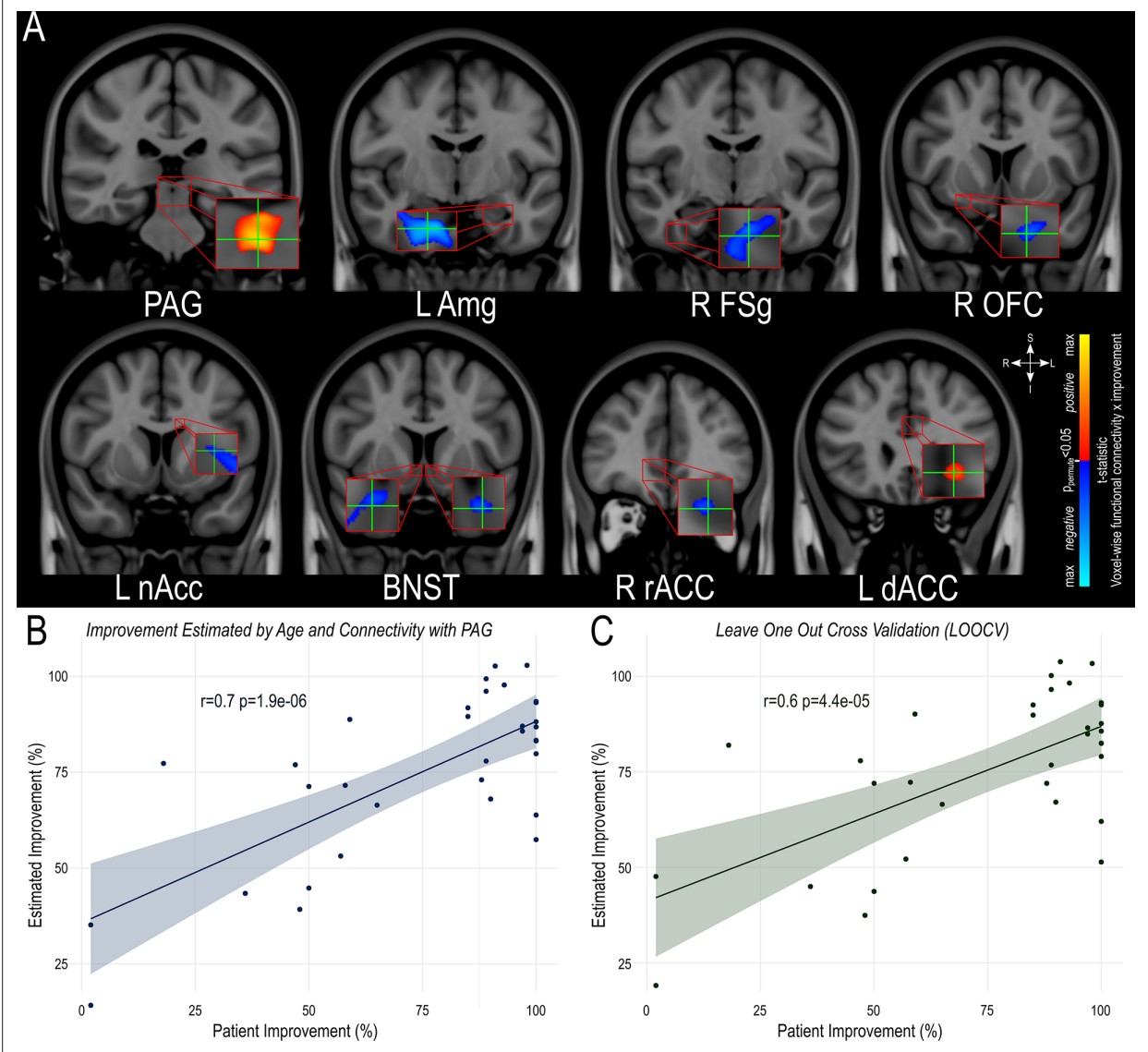

**Figure 6.** Estimation of clinical outcome. (**A**) Location of the peak extracted for each area found to have significant functional connectivity with the volume of activated tissue Illustrated in the coronal plane in MNI152 standard-space (http://www.bic.mni.mcgill.ca/ServicesAtlases/HomePage). (**B**) A model using age and individual VAT connectivity to the periaqueductal gray significantly estimated individual symptom improvement in the whole dataset and (**C**) retained significance during leave-one-out cross-validation (LOOCV).

*Gallagher et al., 2019*, it is possible that treating dysfunctional circuitries while they are still being developed could induce beneficial and long-lasting plastic changes to restore functional normality. DBS is primarily used for the treatment of neurological and psychiatric disorders in adults and is emerging as an effective and safe therapy for neurological diseases in children *Lipsman et al., 2010* such as childhood dystonia *Elkaim et al., 2018*, drug-resistant epilepsy *Yan et al., 2018* and Gilles de la Tourette syndrome *Coulombe et al., 2018*.

Using patients' pre and postoperative imaging, we performed a Voxel Efficacy Mapping investigation and were able to determine a specific area that, when stimulated, is associated with an improvement greater than 90% located in the more posterior-inferior-lateral aspect of the posterior hypothalamic area (coordinates are provided in Talairach-Tournoux space and MNI152 space in the Results section). In humans, DBS and lesions of the posterior hypothalamus have been consistently shown to reduce aggressive symptoms (for a detailed review, see *Gouveia et al., 2019*). One of the most commonly targeted areas in those series lies along the midpoint of the line between the anterior and posterior commissures (AC-PC line), in a region anterior to the rostral end of the aqueduct,

**Table 2.** Estimation of clinical outcome based on functional connectivity map and patient age.

| Functionally connected brain area | Peak coordinate | R | R² | p-value |
|---|---|---|---|---|
| Periaqueductal Grey Matter | x=-1 y=-30 z=-10 | 0.725 | 0.525 | 1.86e-06 |
| Vermis | x=1 y=-49 z=-12 | 0.702 | 0.493 | 5.21e-06 |
| Medial Raphe nucleus | x=0 y=-25 z=-15 | 0.689 | 0.475 | 9.11e-06 |
| Right Subst. Nigra, Subthalamic n., Zona Incerta | x=16 y=-14 z=-7 | 0.681 | 0.464 | 1.28e-05 |
| Left Subst. Nigra, Subthalamic n., Zona Incerta | x=-14 y=-16 z=-7 | 0.673 | 0.453 | 1.77e-05 |
| Left Claustrum | x=-30 y=14 z=-2 | 0.672 | 0.451 | 1.88e-05 |
| Left Amygdala | x=-25 y=-8 z=-27 | 0.672 | 0.451 | 1.88e-05 |
| Right Fusiform Gyrus | x=37 y=-10 z=-34 | 0.668 | 0.447 | 2.14e-05 |
| Left Putamen | x=-33 y=-4 z=2 | 0.656 | 0.430 | 3.41e-05 |
| Left Dorsal Anterior Cingulate Cortex | x=-2 y=25 z=22 | 0.654 | 0.428 | 3.65e-05 |
| Right Superior Parietal Lobule | x=23 y=-62 z=63 | 0.652 | 0.425 | 3.95e-05 |
| Left Transition Orbitofrontal Cortex- Insula | x=-24 y=11 z=-18 | 0.645 | 0.416 | 5.06e-05 |
| Right Nucleus Accumbens | x=11 y=8 z=-7 | 0.645 | 0.416 | 5.07e-05 |
| Right Amygdala | x=-22 y=-6 z=-26 | 0.645 | 0.416 | 5.14e-05 |
| Left Anterior Insula | x=-39 y=18 z=-1 | 0.642 | 0.413 | 5.59e-05 |
| Right Rostral Anterior Cingulate Cortex | x=9 y=40 z=-5 | 0.638 | 0.407 | 6.57e-05 |
| Right Bed Nucleus Of The Stria Terminallis | x=7 y=8 z=-5 | 0.638 | 0.407 | 6.57e-05 |
| Left Bed Nucleus Of The Stria Terminallis | x=-6 y=8 z=-5 | 0.638 | 0.407 | 6.57e-05 |
| Left Nucleus Acuumbens | x=-11 y=8 z=-7 | 0.638 | 0.407 | 6.57e-05 |
| Right Transition Orbitofrontal Cortex- Insula | x=25 y=10 z=-15 | 0.638 | 0.407 | 6.57e-05 |
| Right Hypothalamus | x=5 y=-3 z=-11 | 0.631 | 0.398 | 8.25e-05 |
| Left Fusiform Gyrus | x=-34 y=-12 z=-33 | 0.629 | 0.396 | 8.72e-05 |
| Left Hypothalamus | x=-4 y=-3 z=-11 | 0.621 | 0.386 | 1.13e-04 |

posterior to the anterior border of the mammillary body and superior to the red nucleus *Sano and Mayanagi, 1988* (*Figure 3—figure supplement 1* shows the location of the sweet-spot in relation to the red nucleus and subthalamic nucleus).

Our connectomics analyses provided insight into the structural and functional networks that underlie the clinical efficacy of pHyp-DBS. As no patient-specific functional or diffusion-weighted imaging scans were available, these analyses were performed using high-resolution, high signal-to-noise ratio images derived from healthy individuals that create an average atlas of the functional and connectivity profiles of the human brain *Gouveia et al., 2021c*; *Germann et al., 2021a*; *Elias et al., 2021*; *Germann et al., 2021b*; *Wang et al., 2021*; *Elias, 2022*. The analysis of functional and structural connectivity via normative datasets is an opportunity to investigate possible underlying mechanisms based on the current knowledge of the typical human brain. Unlike patient-specific studies, where functional MRI and diffusion-weighted images are acquired from a single patient, normative analyses rely on high-quality atlases derived from very large cohorts of healthy subjects (around 1000 brain scans). Thus, these exploratory data-driven approaches are intended to investigate the pattern of brain activity and structural connectivity related to a specific region of interest (i.e. individual VAT) and correlate these data with clinical outcomes to simulate the presumed brain changes associated with DBS treatment. This approach allows for the expansion of the investigation to a large, previously inaccessible cohort of patients whose individual data are available, generating knowledge that can help optimize treatment in future patient populations. Although normative connectivity data may not

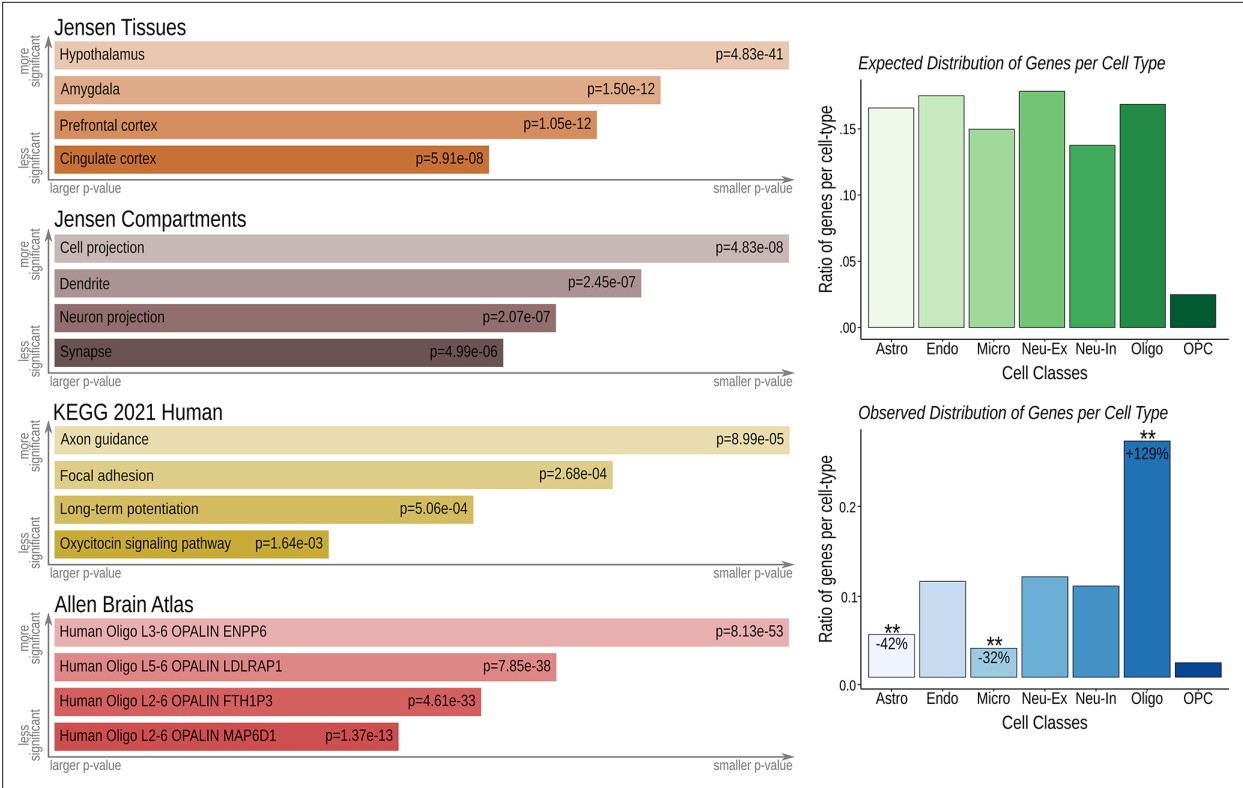

**Figure 7.** Imaging transcriptomics-gene set analysis. The gene set analysis was performed using the TFCE-corrected distribution of clinically relevant functionally connected areas ($q_{FDR}$<0.0001) along with whole brain three-dimensional expression patterns provided by the Allen Brain Atlas (http://human.brain-map.org/) *Hawrylycz et al., 2012*; *Sunkin et al., 2013*; *Shen et al., 2012* averaged into the Harvard-Oxford Atlas (http://www.cma.mgh.harvard.edu/). Genes with similar spatial pattern of distribution to the functional connectivity map were Bonferroni corrected at p<0.005 and selected for further gene ontology analysis. Left panel: The EnRichr tool (https://maayanlab.cloud/Enrichr/) *Kuleshov et al., 2016* was used to investigate associated biological processes, followed by specific tissue and compartment analysis provided by the Jensen Gene Ontology Tool (https://jensenlab.org/resources/proteomics/), the Kyoto Encyclopedia of Genes and Genomes (KEEG; https://www.genome.jp/kegg/) and the Allen Human Brain Atlas (http://human.brain-map.org/). Right panel: A cell-specific aggregate gene set provided by *Seidlitz et al., 2020* was used to determine the cell types associated with these genes. Results were confirmed to be non-random using permutation testing (1000 permutations, ** p<0.01).

fully capture patient- or pathology-specific variations, they have been previously shown to generate results that are comparable to patient-specific imaging data *Elias, 2022*; *Wang et al., 2021*.

Structural connectivity analysis in our study implicated fiber tracts involved in somatosensation (e.g. medial lemniscus and spinothalamic tract), emotional regulation (e.g. amygdalofugal pathway and medial forebrain bundle), and motor signaling (e.g. motor projections and central tegmental tract) in the amelioration of aggressive behavior. Previous studies have shown a significant correlation between aggressive behavior and motor agitation *Gouveia et al., 2020*; *Gouveia et al., 2021b*, with a concomitant reduction in both following neuromodulatory treatments targeting the posterior hypothalamus *Gouveia et al., 2021c*; *López Ríos et al., 2023*. Indeed, patients with neurodevelopmental disorders, especially those presenting with self-injurious behavior, are known to frequently have altered sensory perception (including reduced pain sensitivity), which can contribute to the chronicity and severity of the behaviour *Gouveia et al., 2021a*; *Koenig et al., 2017*. Furthermore, self-injurious behaviors in this patient population are considered to be a type of repetitive/stereotypic behavior that results in physical injury to the patient's own body *Hagopian et al., 2015*; *Yan et al., 2022*, highlighting the relevance of the motor system in the network subserving aggressive behavior. As such, modulation of somatosensory and motor pathways – perhaps leading to increased pain perception or awareness, and reduction in repetitive/stereotypic behaviors – could help reduce self-injurious behaviors.

More generally, aggressive behaviors and motor agitation are believed to occur as a result of a decreased tolerance to provocative stimuli caused by reduced serotonergic transmission in the

prefrontal cortex, resulting in an ineffective top-down inhibitory control over a consequently hyper-activated amygdala. The amygdala, in turn, sends alert signs to the periaqueductal gray and hypothal-amus for motor activation and hormonal production, preparing the organism for a response *Gouveia et al., 2021a*; *Gouveia et al., 2019*; *Gouveia et al., 2020*; *Gouveia et al., 2021b*; *Blair, 2016*. Thus, it is conceivable that stimulation of fiber tracts involved in emotional regulation and motor outputs could mediate clinical improvement by reducing motor agitation, improving concentration and learning, and increasing tolerance to provocative external stimuli. The functional connectivity mapping showed that the successful VATs were functionally connected to several key areas within the neurocircuitry of aggressive behavior, including the amygdala, nucleus accumbens, hypothalamus, periaqueductal gray, and cingulate cortex. To further explore these results, we investigated the relationship between the functional connectivity of each of these brain structures and individual treatment outcomes and observed that functional connectivity with the PAG is the most relevant. In addition, when modeling the functional connectivity of any two brain areas identified in the functional connectivity analysis, we observed that the most relevant connections involved the PAG and limbic structures believed to be at the core of the neurocircuitry of aggressive behavior, such as the amygdala, anterior cingu-late cortex, and nucleus accumbens. In line with our findings, previous studies have shown positive outcomes following ablative and neuromodulation treatments targeting the amygdala, hypothalamus, and nucleus accumbens for the control of aggressive behavior *Gouveia et al., 2021a*; *Gouveia et al., 2019*; *Gouveia et al., 2021b*; *Yan et al., 2022*. Moreover, PAG DBS – primarily performed to treat neuropathic pain *Knotkova et al., 2021* – has been reported to induce changes in mood and anxiety *Gray et al., 2014*.

Another aspect detected by the functional connectivity analysis was the involvement of areas responsible for the production of serotonin (i.e. raphe nuclei) and dopamine (i.e. substantia nigra). Both these monoamines are thought to be directly involved in the initiation and maintenance of aggressive behaviors due to their role in facilitating top-down inhibitory controls, regulating mood, and mediating social behaviors *Gouveia et al., 2019*; *Gouveia et al., 2021c*; *Miczek et al., 2002*. Antipsychotics (both typical and atypical) and selective serotonin reuptake inhibitors – the most commonly used drugs to control of aggressive behaviors – act on the dopaminergic and seroto-nergic neurotransmission systems *Gouveia et al., 2021a*; *Gouveia et al., 2019*. In a mouse model of escalated aggression, pHyp-DBS increased the density of serotonin receptors type 1A (5-HT1$_A$) in the orbitofrontal cortex and amygdala, while pretreatment with a 5-HT1$_A$ antagonist blocked the anti-aggressive effect of stimulation *Gouveia et al., 2023*.

To further explore the dataset pooled in our study, we performed a spatial imaging transcrip-tomic analysis. Our goal was to investigate spatial correlations between gene expression patterns (i.e. microarray data from the Allen Human Brain Atlas) and imaging characteristics (i.e. functional connec-tivity map derived from the patient's VATs) and possible molecular characteristics associated with symptom improvement (for a review on imaging transcriptomics see *Fornito et al., 2019*). Imaging transcriptomics is a powerful non-invasive method to measure neural phenotypes and investigate neurobiological mechanisms of disease or treatments *Arnatkeviciute et al., 2022*. Genes whose spatial pattern correlates with brain changes in imaging transcriptomics are also found in post-mortem and large-scale GWAS studies of specific patient populations *Arnatkeviciute et al., 2022*. In fact, several studies have demonstrated associations between the spatial distribution of genes and neuro-developmental changes *Xie et al., 2020*; *Romero-Garcia et al., 2019* as well as psychiatric disor-ders *Romero-Garcia et al., 2020*. In this study, we identified a number of genes with similar spatial distribution to brain regions that showed clinically important functional connectivity with the stimu-lation target. The biological processes associated with these genes were investigated using several gene ontology analysis tools. The Jensen Tissues library revealed a list of brain structures classically associated with aggressive behavior, including those associated with the limbic system (e.g. amyg-dala and hypothalamus) and those involved in higher-order cognitive centers (e.g. prefrontal cortex). As described above, dysregulation in the top-down inhibitory control of the prefrontal cortex over the limbic system is thought to subserve aggressive behaviors. Additionally, the KEGG 2021 Human Library identified genes implicated in the oxytocin signaling pathway, a peptide hormone critically implicated in aggressive behavior *Neumann, 2008*. It has been shown that intranasal administration of oxytocin increases human aggressive responses when subjects are evaluated in a social orientation paradigm *Ne'eman et al., 2016*. Moreover, single-nucleotide polymorphisms in the oxytocin gene

have been associated with childhood-onset aggressive behaviors *Malik et al., 2012* and with aggressive behaviors in men intoxicated with alcohol *Johansson et al., 2012*.

A large number of genes identified by the Jensen Compartments and KEGG 2021 Human Libraries are part of biological processes associated with neuronal communication and plasticity, thus possibly linked to the intrinsic mechanism of action of DBS *Hamani and Temel, 2012*; *Florence et al., 2016*. As a neuromodulation therapy, DBS induces long-lasting changes in cellular and molecular aspects of neurons belonging to dysfunctional neurocircuitries in order to restore functional normality *Jakobs et al., 2019*; *Karas et al., 2013*; *Lozano et al., 2019*. Although its neurobiological mechanisms are not fully understood, DBS is believed to exert its effects by altering the cellular membrane potential, resulting in either an increase or a decrease in action potentials *Jakobs et al., 2019*; *Karas et al., 2013*; *Lozano et al., 2019*. These changes are then propagated throughout neural networks, changing neurotransmitter dynamics, protein expression, and membrane receptor availability *Jakobs et al., 2019*; *Karas et al., 2013*; *Lozano et al., 2019*. In fact, a previous study using imaging transcriptomics to investigate brain changes after functional neurosurgery of the amygdala in patients with refractory aggressive behavior has also identified several genes related to neuronal communication and plasticity *Gouveia et al., 2021a*. Preclinical studies have also found significant increases in neural precursor cells, plasticity, and precursor cell markers in animals implanted with DBS, along with significant reductions in the number of activated microglia and astrocytes *Vedam-Mai et al., 2016*; *Leplus et al., 2019*; *Chan et al., 2018*; *Bambico et al., 2015*; *Encinas et al., 2011*. This is in line with the cell type investigation, where a significant reduction in the ratio of genes per cell type was observed for astrocytes and microglia when compared to the expected distribution. Both astrocytes and microglia are highly implicated in mechanisms of neuroinflammation *Carson et al., 2006*, which in turn is associated with several psychiatric disorders *Najjar et al., 2013*. The reduction of neuroinflammatory markers (such as glial activation and interleukin levels) has been described in several preclinical DBS studies and is thought to be involved in the mechanism of action of DBS *Chan et al., 2018*; *Chen et al., 2020*; *Amorim et al., 2015*.

Some limitations of this study are of note; with the exception of the analysis of the most efficacious location of stimulation (i.e. voxel efficacy map), which is fully derived from patient data, the remaining exploratory analyses were performed using normative datasets. This methodology was chosen as neither patient-specific functional nor diffusion-weighted imaging was available. Sophisticated imaging acquisitions, such as the ones necessary for the connectivity analysis, are not frequently acquired in patients with implanted DBS, as many devices have not been considered safe for MRI use. Additionally, it is frequent for health centers to have a 'no-MRI' policy for patients with implanted DBS, only allowing for the acquisition of CTs following surgery. It is important to highlight that previous studies showed that normative datasets could be used to create probabilistic models of optimal connectivity associated with patients' outcomes that are meaningful to predict outcomes in patient-specific connectivity data *Wang et al., 2021*. Similarly, individual brain genetic profiles were not available for this patient population. The imaging transcriptomic analysis performed here was based on the Allen Human Brain Atlas, which is derived from post-mortem biopsies of multiple adult control brains *Hawrylycz et al., 2012*; *Arnatkeviciute et al., 2019*; *Arnatkeviciute et al., 2023*; *Markello et al., 2021*, and has been shown to be relevant to large-scale GWAS studies of specific patient populations *Arnatkeviciute et al., 2022*.

The structural connectivity analysis performed here provides a detailed overview of the relevant tracts that are related to symptom alleviation in this patient population. However, for precise segregation of these fiber tracts and the determination of those that are necessary for symptom improvement, it would be necessary to perform preclinical experiments to selectively activate/inactivate specific fiber tracts and investigate the impact of these changes on behavior and network dynamics. Nevertheless, understanding how the location of the most efficacious area of stimulation correlates with these tracts is important for refining the surgical planning of future patients, as one could use individual patient tractography to determine the optimal location of the electrodes based on how it would impact individual fiber tracts.

## Conclusion

This integrated imaging analysis of the largest multi-center international dataset of patients treated with posterior hypothalamus deep brain stimulation (pHyp-DBS) for refractory and severe aggressive

behavior showed that treatment efficacy is high, with over 90% of patients presenting significant symptom improvement. The probabilistic sweet-spot mapping revealed an optimized surgical target where maximal symptom alleviation could be achieved, and the stereotaxic coordinates of this optimal stimulation site are provided, allowing for direct clinical application. The connectomics analysis performed here expands our understanding of the neurocircuitry of aggressive behavior and suggests that engagement of specific brain areas within the neurocircuitry is key for symptom alleviation. Further studies are necessary to investigate if neurostimulation of these deep brain structures with non-invasive neuromodulatory treatments, such as deep transcranial magnetic stimulation (dTMS) or low-intensity focused ultrasound (LIFUS), would result in beneficial outcomes for patients.

## Methods

### Patients included

Thirty-three patients from five international centers were included in this study: Hospital Universitario San Vicente Fundación, Colombia; University Hospital La Princesa, Spain; International Misericordia Clinic, Colombia; Universidad Autónoma de Bucaramanga and FOSCAL Clinic, Colombia; Sírio-Libanês Hospital, Brazil. Detailed demographics, medical history, surgical procedure, and relevant information regarding institutional ethical review board and informed consent can be found in previous publications *Gouveia et al., 2021c*; *Contreras Lopez et al., 2021*; *Benedetti-Isaac et al., 2015*; *Torres et al., 2020*; *Micieli et al., 2017*. All patients had pre-operative T1w MRI images as well as postoperative imaging (CT or MRI) that allowed for the precise determination of individual electrode placement (imaging parameters may be found in the original publications of the individual centers) *Gouveia et al., 2021c*; *Contreras Lopez et al., 2021*; *Benedetti-Isaac et al., 2015*; *Torres et al., 2020*; *Micieli et al., 2017*.

### Electrode localization and volume of activated tissue (VAT) modeling

Lead-DBS software (https://www.lead-dbs.org/) was used for electrode localization and modeling of the volume of activated tissue, as previously described *Horn and Kühn, 2015*. Following intensity inhomogeneity correction, the postoperative MRI or CT and the preoperative MRI scans were rigidly co-registered using SPM12 (https://www.fil.ion.ucl.ac.uk/spm/software/spm12/ *Penny et al., 2011*). Images were non-linearly normalized to standard space (ICBM 2009b NLIN asymmetric) using Effective Low Variance ANTS (http://stnava.github.io/ANTs), and an additional subcortical affine transformation was implemented to correct for post-operative brain shift *Avants et al., 2011*. Electrodes located in the right and/or left hemispheres were manually localized using the post-operative scan and then warped to standard space using the previously generated transforms. The patient's unilateral or bilateral VATs associated with individual stimulation parameters at the last follow-up (individual stimulation parameters are described in *Table 1*) were modeled by first constructing a four-compartment volume conductor model (http://iso2mesh.sourceforge.net/cgi-bin/index.cgi) that segregated peri-electrode tissue by tissue type *Ewert et al., 2018*. Next, the FieldTrip-SimBio finite element model pipeline was used to simulate the potential electric field distribution around each active contact (https://www.mrt.uni-jena.de/simbio/index.php/; http://fieldtriptoolbox.org) and binary VATs were generated by thresholding the gradient of this distribution at 0.2 V/mm. Considering that increased pulse width increases the effective range of stimulation (thus translating to a larger VAT) by lowering the activation threshold of nearby axons *Duffley et al., 2019*, and given that our patient cohort included a small number of patients who received higher pulse widths than the ones assumed by the FieldTripSimBio model in Lead-DBS, we performed a supplementary analysis to investigate changes in VAT volumes by also modeling the pulse width observed in these patients. To this aim, we used a simpler heuristic model *Dembek et al., 2017* that takes pulse width into account to compute additional VATs using pulse width values ranging from 90µs up to the patient's actual pulse width values. As expected, this analysis yielded larger VATs when higher pulse width values were used, with an absolute difference in VAT diameter between 90 µs and 450 µs (the highest pulse width observed in this cohort) of 2 mm. Then, we investigated whether or not these larger VATs could have potentially impacted our results by performing a new probabilistic mapping analysis using the newly generated VATs (specifically, the largest VATs that were enlarged by 2 mm in diameter) for the patients with higher pulse width values. This new analysis yielded a very similar average map to the original analysis, with the overall

map pattern and location/values of the peak corresponding to the most efficacious area for maximal symptom alleviation remaining unaltered with only a few voxels on the periphery of the map changing in value by a couple of percentage points (*Figure 3—figure supplement 2*). As such, this supplementary analysis indicates that our results were not meaningfully altered by the unusual pulse width observed in these patients.

## Voxel efficacy mapping

Probabilistic maps of efficacious voxels were generated to provide insight into spatial patterns of response to pHyp-DBS, as previously described *Elias et al., 2021*; *Dembek et al., 2017*; *Germann et al., 2021b*. All VATs were flipped to the right hemisphere, and each VAT was weighted by the percentage of improvement from the baseline. Mean improvement at each voxel was computed by averaging the normalized weighting values. The resultant raw average map was then masked by a frequency map thresholded at 10% in order to exclude outlier voxels. Finally, a statistical map thresholded at $p < 0.05$ was calculated using the Wilcoxon signed-rank test to determine whether pHyp-DBS was associated with a significant difference in clinical change at each voxel. The validity of the voxel efficacy map was confirmed using a non-parametric permutation analysis, in which each clinical score was randomly assigned to a random VAT. The voxel map was determined to be significant ($p_{permute} < 0.01$). The coordinates relative to the most efficacious area of stimulation were extracted in MNI space and converted to Talairach space using the MNI-Talairach Converter from BioImage Suite Web (https://bioimagesuiteweb.github.io/webapp/mni2tal.html) *Lacadie et al., 2008*. All imaging analyses were performed using the MNI152 non-linear template (https://www.bic.mni.mcgill.ca/ServicesAtlases/ICBM152NLin2009).

## Imaging connectomics analyses - structural and functional connectivity mapping

The brain-wide imaging connectomics analyses of the patient's unilateral or bilateral VATs were explored using high-quality Structural (diffusion MRI-based tractography derived streamlines) and Functional (resting-state functional MRI-derived voxel-wise functional pattern) normative connectomes, as previously described *Elias et al., 2021*; *Elias et al., 2020*; *Germann et al., 2021b*; *Coblentz et al., 2021*. Briefly, for each patient's individual VAT(s), a whole-brain r-map was computed based on the resting-state functional MRI BOLD time course-dependent correlations between the seed region and the remaining voxels in the brain (the processed rsfMRI connectome data and script are freely available through lead-DBS https://www.lead-dbs.org. In-house MATLAB script, The MathWorks, Inc, Version R2017b. Natick, MA, USA) across 1000 healthy subjects (Brain Genomics Superstruct Project dataset, age range: 18–35 years; 57% female, http://neuroinformatics.harvard.edu/gsp) *Fox et al., 2014*. To exclude voxels with potentially spurious correlations, all individual r-map were corrected for multiple comparisons by converting them to a t-map and thresholded at $p_{corrected} < 0.05$ (whole-brain voxel-wise Bonferroni correction) using the known p-distribution *Fox, 2018*; *Mansouri et al., 2020*. Thus, only voxels significantly functionally connected to each patient's VATs were used for the subsequent analysis. For generating group-level maps, a voxel-wise linear regression analysis investigating the relationship between the functional connectivity of the VATs and the individual clinical outcome was then performed, followed by non-parametric permutation analysis. The individual improvement scores were randomly assigned to a functional connectivity map, and the analysis was repeated 1000 times. At each voxel, $p_{permute} < 0.05$ was used as a significance threshold for the final functional connectivity analysis. *Figure 1—figure supplement 1* illustrates this process.

Structural connectivity outputs were obtained by identifying all streamlines that touched the uni or bilateral VATs (used as seeds) out of an approximately 12 million-fibers whole-brain tractography template (the processed dMRI connectome data and script are also freely available through lead-DBS https://www.lead-dbs.org. In-house MATLAB script, The MathWorks, Inc, Version R2017b. Natick, MA, USA). This template was assembled from a 985-subject multi-shell diffusion-weighted MRI Human Connectome Project dataset (http://www.humanconnectomeproject.org) using generalized q-sampling imaging (http://dsi-41studio.labsolver.org/). To determine what streamlines are significantly associated with the outcome, a t-test of symptom improvement comparing subjects whose VAT touched a given streamline and individuals who did not was performed as previously described *Li*

*et al., 2020*; *Germann et al., 2021b*. The results were corrected for multiple comparisons using False Discovery Rate (FDR) at $q_{FDRcor}$ <0.001.

## Imaging transcriptomics - gene set analysis

We performed a gene set analysis using the abagen toolbox (https://abagen.readthedocs.io/en/stable/index.html) to investigate genes whose spatial pattern distribution is similar to the pattern of clinically relevant functional connectivity. For this analysis, we used the Allen Human Brain Atlas (https://alleninstitute.org/) microarray data describing the cortical, subcortical, brainstem and cerebellar localization of over 20,000 genes in the human brain (3702 anatomical locations from 6 neurotypical adult brains) *Hawrylycz et al., 2012*; *Arnatkeviciute et al., 2019*; *Arnatkeviciute et al., 2023*; *Markello et al., 2021*, along with a cell-specific aggregate gene set *Seidlitz et al., 2020*. These data are provided preprocessed, with gene expression values normalized across all donors' brains, and registered to standard MNI space allowing for the direct comparison between the spatial pattern of gene expression and the functional connectivity map (https://human.brain-map.org/microarray/search) *Arnatkeviciute et al., 2019*. To achieve this spatial mapping, we first applied Threshold Free Cluster Enhancement (TFCE) to the functional connectivity map to generate clusters of significant functional connectivity with a spatial extent that overlaps with the anatomical locations where microarray data was obtained. Then, based on the Harvard-Oxford Atlas (http://www.cma.mgh.harvard.edu/), a regional parcellation covering 48 cortical and 21 subcortical areas in each hemisphere (a total of 138 parcellations) was applied to both datasets. A direct correlation between the spatial distribution of gene expression and functional connectivity analysis in each specific brain area was performed and corrected for multiple comparisons using FDR at $q_{FDAcor}$ <0.005, resulting in a gene list composed of 1734 genes associated with biological processes and ratios of genes per cell type. This list was used as input for gene ontology analyses using the EnRichr Gene Ontology tool (https://maayanlab.cloud/Enrichr/), along with the Jensen Gene Ontology enrichment tool (https://jensenlab.org/resources/proteomics/) and the Kyoto Encyclopedia of Genes and Genomes (KEGG pathway enrichment; https://www.genome.jp/kegg/kegg1b.html), resulting in an extensive list of associated biological terms that were corrected for multiple comparisons using FDR at $q_{FDAcor}$ < 0.005.

## Statistical analysis

R (version 3.4.4; https://www.r-project.org/) was used for statistical analysis. The RMINC package (https://github.com/Mouse-Imaging-Centre/RMINC; *Mouse Imaging Centre, Hospital for Sick Children, Toronto Canada, 2022*) was used for the analysis of imaging data. Linear models were used to investigate the percentage of improvement by age, and t-tests were performed to investigate possible differences in symptom improvement comparing male and female patients. Additive linear models were used to test our ability to estimate the percentage of improvement from baseline using age and the functional connectivity of all brain areas functionally connected with the VATs. The level of significance for all analyses was set at a minimum of $p < 0.05$.

## Study approval

Individual trials and cases were evaluated by the corresponding local ethics committees. Written informed consent was obtained. Five international centers shared clinical data for this study. 1. Comité de Ética de la Investigación of Hospital Universitario San Vicente Fundación (#08–2022). 2. Comité de Ética de la Investigación con Medicamentos of Hospital Universitario La Princesa. 3. Comité de Ética de los Estudios Clínicos of La Misericordia Clínica Internacional (2012). 4. Comité Institucional de Ética of Universidad Autónoma de Bucaramanga. 5. Comitê de Ética em Pesquisa of Sociedade Beneficente de Senhoras Hospital Sírio-Libanês (#27470619.8.0000.5461).

## Acknowledgements

This work was supported by funds from the Harquail Center for Neuromodulation, Canadian Institutes of Health Research (CIHR) Postdoctoral Fellowship #472484 (FVG), Fundacao de Amparo a Pesquisa do Estado de Sao Paulo (FAPESP) #13/20602–5 (FVG), Fundacao de Amparo a Pesquisa do Estado de Sao Paulo (FAPESP) #17/10466–8 (FVG), Canadian Institutes of Health Research (CIHR) Banting fellowship #471913 (JG), and Fundacao de Amparo a Pesquisa do Estado de Sao Paulo (FAPESP) #11/08575–7 (RCRM). The authors wish to thank the research assistants and staff of Hospital

Sirio-Libanes, Brazil; Hospital Universitario San Vicente Fundación, Colombia; University Hospital La Princesa, Spain; International Misericordia Clinic, Colombia; Universidad Autónoma de Bucaramanga and FOSCAL Clinic, Colombia.

## Additional information

### Funding

| Funder | Grant reference number | Author |
|---|---|---|
| Canadian Institutes for Health Research | 72484 | Flavia Venetucci Gouveia |
| Fundacao de Amparo a Pesquisa do Estado de Sao Paulo | 13/20602-5 | Flavia Venetucci Gouveia |
| Fundacao de Amparo a Pesquisa do Estado de Sao Paulo | 17/10466-8 | Flavia Venetucci Gouveia |
| Canadian Institutes for Health Research | 471913 | Jurgen Germann |
| Fundacao de Amparo a Pesquisa do Estado de Sao Paulo | 11/08575-7 | Raquel Chacon Ruiz Martinez |

The funders had no role in study design, data collection and interpretation, or the decision to submit the work for publication.

### Author contributions

Flavia Venetucci Gouveia, Conceptualization, Data curation, Formal analysis, Funding acquisition, Investigation, Methodology, Writing – original draft, Project administration, Writing – review and editing; Jurgen Germann, Conceptualization, Data curation, Formal analysis, Funding acquisition, Investigation, Methodology, Writing – original draft, Writing – review and editing; Gavin JB Elias, Formal analysis, Methodology, Writing – review and editing; Alexandre Boutet, Aaron Loh, Peter Giacobbe, Han Yan, George M Ibrahim, Nir Lipsman, Andres Lozano, Writing – review and editing; Adriana Lucia Lopez Rios, Cristina Torres Diaz, William Omar Contreras Lopez, Raquel Chacon Ruiz Martinez, Erich Talamoni Fonoff, Juan Carlos Benedetti-Isaac, Pablo M Arango Pava, Data curation, Writing – review and editing; Clement Hamani, Conceptualization, Funding acquisition, Writing – original draft, Writing – review and editing

### Author ORCIDs

Flavia Venetucci Gouveia (iD) http://orcid.org/0000-0003-0029-1269
Jurgen Germann (iD) http://orcid.org/0000-0003-0995-8226

### Ethics

Individual trials and cases were evaluated by the corresponding local ethics committees. Written informed consent was obtained. Five international centers shared clinical data for this study. 1. Comité de Ética de la Investigación of Hospital Universitario San Vicente Fundación (#08-2022). 2. Comité de Ética de la Investigación con Medicamentos of Hospital Universitario La Princesa. 3. Comité de Ética de los Estudios Clínicos of La Misericordia Clínica Internacional (2012). 4. Comité Institucional de Ética of Universidad Autónoma de Bucaramanga. 5. Comitê de Ética em Pesquisa of Sociedade Beneficente de Senhoras Hospital Sírio-Libanês (#27470619.8.0000.5461).

### Decision letter and Author response

Decision letter https://doi.org/10.7554/eLife.84566.sa1
Author response https://doi.org/10.7554/eLife.84566.sa2

## Additional files

**Supplementary files**
• Supplementary file 1. Matrix of the correlations between estimated symptom improvement (i.e. linear model of age and functional connectivity of the two areas) and the measured improvement.
• MDAR checklist

**Data availability**
The codes for electrode localization, modelling of the volume of activated tissue, and imaging connectomics (i.e. functional and structural connectivity) are freely available in Lead-DBS (https://www.lead-dbs.org/). The codes, along with the Allen Human Brain Atlas (AHBA) microarray dataset, for the analysis of spatial transcriptomics are freely available in abagen (https://abagen.readthedocs.io/en/stable/). Along with the codes, the websites for these two toolboxes provide manuals describing the step-by-step procedure for successful analysis. The dataset accompanying this study is freely available at Zenodo (https://doi.org/10.5281/zenodo.7344268).

The following previously published dataset was used:

| Author(s) | Year | Dataset title | Dataset URL | Database and Identifier |
|---|---|---|---|---|
| Gouveia FV, Jurgen G | 2022 | Dataset accompanying this study | https://doi.org/10.5281/zenodo.7344268 | Zenodo, 10.5281/zenodo.7344268 |

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
