## [Editor Report]

This study presents useful structural and functional connectivity profiles of patients receiving deep brain stimulation in the posterior hypothalamus for severe and refractory aggressive behavior. The inclusion of data from multiple centers is compelling. This study will be important for a broad readership including basic and clinical neuroscientists.

---

## [Decision Letter]

**Decision letter after peer review:**

Thank you for submitting your article "Networks and genes modulated by posterior hypothalamic stimulation in patients with aggressive behaviours: Analysis of probabilistic mapping, normative connectomics, and atlas-derived transcriptomics of the largest international multi-centre dataset" for consideration by *eLife*. Your article has been reviewed by 2 peer reviewers, and the evaluation has been overseen by a Reviewing Editor and Michael Frank as the Senior Editor. The reviewers have opted to remain anonymous.

Essential revisions:

1. The validation of findings is heterogeneous and inconsistent across analysis pipelines. While the authors performed non-parametric permutation testing during sweet-spot mapping, structural and functional connectivity were validated using a 'four-fold consistency analysis'. The latter consists of a visual representation of streamlines and peak intensities after randomly dividing data into four groups, the findings were not validated quantitatively. If possible, the authors should apply permutation analysis in alignment with sweet-spot mapping and demonstrate the predictive ability of their identified networks in a LOO or k-fold cross-validation paradigm as carried out by similar studies. Given that the data has been derived from multiple centers, the prediction of left-out cohorts based on models generated by the remaining cohorts could be another means of validation. If validation is not possible, the authors should clearly state the limitations of their approach.

2. In addition to a 'four-fold consistency analysis', functional connectivity was evaluated using LOOCV in a-priori-identified ROIs. Their network analysis, however, revealed a far more extensive network encompassing cortical, subcortical, and cerebellar structures. To avoid selection bias the authors should incorporate identified structures into their analysis and apply appropriate means of validation.

3. Functional connectivity mapping: how were R-maps generated? The authors mention that patient-specific R-maps were p-thresholded and corrected for multiple comparisons, but it is not clear how group-level maps were generated. How did the authors perform regression on these maps? Were voxels that did not survive thresholding excluded?

4. The authors determined that age was a significant prédictor of the outcome, but it is unclear whether certain age groups presented with distinct etiologies underlying their aggressiveness. For example, aggression in epilepsy may show a better response to DBS as opposed to schizophrenia. How does patient outcome change when stratifying according to etiology? How does model performance change when controlling for etiology? The authors should include the etiology of aggressiveness in Table 1.

5. Stimulation parameters. The authors report average pulse widths of 219 µs and 142µs respectively, which is up to 4-fold higher as compared to DBS settings used conventionally in movement disorders and will significantly alter the volume of activated tissue. Did the authors account for the drastic increases in pulse width during VAT modeling?

6. Imaging transcriptomics. The methods described lack detail: How did the authors account for differences in expression across donors, samples, and regions during preprocessing of the Allen Human Brain Atlas? How was expression data collapsed into regions of interest? Did the authors apply any normalization? Recent publications have introduced reproducible workflows for processing and preparing the AHBA expression data for analysis that is publicly available.

7. 'genes with similar patterns of spatial distribution to the TFCE map were compiled in an extensive list'. It is unclear why authors used TFCE maps for spatial transcriptomics as opposed to the functional connectivity map featured in Figure 5. How was similarity measured between the TFCE map and the AHBA? How were candidate genes identified? Please provide a more comprehensive description of the analysis pipeline.

8. What do the bar plots in Figure 7 (left) represent? P-values? The authors should label the axes to make this clear to the reader.

9. Interprétation of imaging transcriptomics: The authors identify a therapeutic circuit associated with deep brain stimulation of the posterior hypothalamic area, however, it is unclear how to reconcile genes associated with hormones, inflammation, and plasticity in this context. The authors mention and discuss genes implicated in hormonal processing, specifically oxytocin. The results provided in Figure 7, however, do not support this finding and it is unclear how the authors identified genes linked to oxytocin. In addition, the authors identified reductions in the number of microglia and astrocytes, while oligodendrocytes were overexpressed relative to the expected distribution of genes per cell type. These findings were attributed to DBS effects, however, both connectomic and transcriptomic data are acquired from healthy subjects, which suggests a physiological deficit/enrichment in a therapeutic circuit. How do the authors interpret findings given that no electrode implantation and stimulation were performed?

10. Data availability. Code used for data processing should be made openly available or shared as source data along with the Figures that were generated using the code. Sweet-spot, structural, and functional connectivity maps should be shared openly.

11. Methods: Given the small number of patients over 5 centers, the authors should better address the similarities and differences of data collection across the sites.

12. Results: Clinically relevant bundles included somatosensory, emotions, and motor connections. This is pretty much a summary of connections through the hypothalamus and surrounding regions. Do the authors suggest that the effectiveness of DBS in this region is this nonspecific?

13. Indeed, it appears that the motor system is most involved based on the tractography. This is consistent with figure 3 in which the 'sweet spot' appears to be lateral to the hypothalamus and more centered in the red nucleus/STN region (Figures 1 and 3).

14. The emphasis on connections on monoamine connections (P. 7) is unwarranted, given the stronger connections to motor regions. Moreover, both DBS and tractography favor myelinated fibers over unmyelinated ones. The monoamines are unmyelinated. Finally, the MFB is a relatively small bundle compared to others in the region that carry motor and sensory information to and from the brainstem. The resolution of tractography is unlikely to distinguish the MFB and favor other fiber tracts.

15. In this regard, the paragraph on P. 9, "functional connectivity…" is confusing. (My understanding is that the authors are linking connections to the amygdala, etc.) via the monoamines. Is that correct? If so, that's a discussion point and not well supported by the connectivity data.

16. Other comments: The introduction lacks clear text referring to the organization and connections of the hypothalamus based on classic literature.

17. Examples. First, the authors described the division into 3 regions: anterior, medial, and posterior (reference is the author's article in neurosurgery). Then, without reference to these three regions, describe preclinical studies of the lateral and ventromedial hypothalamus. How do these two areas relate to the three regions?

---

## [Author Response]

Essential revisions:1) The validation of findings is heterogeneous and inconsistent across analysis pipelines. While the authors performed non-parametric permutation testing during sweet-spot mapping, structural and functional connectivity were validated using a 'four-fold consistency analysis'. The latter consists of a visual representation of streamlines and peak intensities after randomly dividing data into four groups, the findings were not validated quantitatively. If possible, the authors should apply permutation analysis in alignment with sweet-spot mapping and demonstrate the predictive ability of their identified networks in a LOO or k-fold cross-validation paradigm as carried out by similar studies. Given that the data has been derived from multiple centers, the prediction of left-out cohorts based on models generated by the remaining cohorts could be another means of validation. If validation is not possible, the authors should clearly state the limitations of their approach.

We agree with the reviewer, and have now improved the validation of our connectomics analyses and removed the four-fold consistency analysis. As for the functional connectivity analysis, we performed a 1000 permutation test (p<0.05), which resulted in an extremely similar map, with the same main brain areas being detected in the corrected and uncorrected maps. Regarding the structural connectivity analysis, we used False Discovery Rate (FDR) correction at a significant level of p<0.001, as it is not feasible to perform a 1000 permutation test with this data. The structural connectome is composed of 12 million fibres, and every single permutation takes approximately 4 hours to be completed using our most powerful computational system. To perform 1000 permutations, it would take at least 4000 hours (i.e. 167 days or 5.5 months) of uninterrupted analysis to complete the test. However, it is important to highlight that an FDR correction at the level of p<0.001 is an extremely stringent method. This means that of the 23,000 fibres detected as being touched by the VATs, only 23 could be incorrect, while the remaining 22,977 are correct. Here again, we observed many similarities between the uncorrected and corrected maps, with the main anatomical structures being detected in both.

We replaced the brain slices in Figures 4 and 5 to demonstrate the new corrected maps and added the information to the Methods section as follows:

Methods, section: Imaging Connectomics Analyses – Structural and Functional Connectivity Mapping. pg 15, 2nd paragraph:

“For generating group-level maps, a voxel-wise linear regression analysis investigating the relationship between the functional connectivity of the VATs and the individual clinical outcome was then performed, followed by non-parametric permutation analysis. The individual improvement scores were randomly assigned to a functional connectivity map, and the analysis was repeated 1000 times. At each voxel, p_permute_<0.05 was used as a significance threshold for the final functional connectivity analysis. Figure 1—figure supplement 1 illustrates this process.”

Methods, section: Imaging Connectomics Analyses – Structural and Functional Connectivity Mapping. pg 17, 1st paragraph:

“The results were corrected for multiple comparisons using False Discovery Rate (FDR) at q_FDRcor_<0.001.”

2) In addition to a 'four-fold consistency analysis', functional connectivity was evaluated using LOOCV in a-priori-identified ROIs. Their network analysis, however, revealed a far more extensive network encompassing cortical, subcortical, and cerebellar structures. To avoid selection bias the authors should incorporate identified structures into their analysis and apply appropriate means of validation.

We would like to thank the reviewer for this valuable suggestion. We originally did not explore the various significant areas but performed a more focused analysis intended to demonstrate that regions of the known ‘aggression network’ are indeed implicated in our findings. We performed a new analysis exploring the correlation between symptom improvement and the functional connectivity of all the areas described in Figure 5 (i.e. functional connectivity map). To this aim, we extracted individual connectivity values from the peak within each significant region and performed the same additive linear model as before, incorporating the functional connectivity of each area, as well as patient age, to estimate individual symptom improvement. In addition, we performed a complete exploratory analysis considering the connectivity of any 2 brain structures and age. The resulting matrix shows to what extent functional connectivity to any two areas can be used to estimate clinical outcomes. Interestingly, this new analysis revealed the Periaqueductal Grey matter (PAG) to be the most important functionally connected area when investigated alone or in combination with brain structures critically involved in the regulation of emotional responses, namely the amygdala, anterior cingulate cortex, bed nucleus of the stria terminalis, nucleus accumbens, orbitofrontal cortex and fusiform gyrus. Also, PAG significance was retained during leave-one-out cross-validation (LOOCV). We added a new Table 2 and Supplementary File 1 showing the results of these analyses, revised Figure 6 and adjusted the text to describe the new analysis and results in the Results, Methods and Discussion sections, as follows:

Results, section: Estimation of Clinical Outcome, pg 8, 2nd paragraph:

“To investigate whether individual functional connectivity to particular hubs within the neurocircuitry of aggressive behaviour could be used to estimate symptom improvement following pHyp-DBS, additive linear models were created. For this, we extracted individual connectivity values from the peak within each brain area where functional connectivity with the VATs was found to be significantly related to outcomes at the group level (as described in the Normative Connectomics Analyses, Figure 6A). The best-performing parsimonious model, which incorporated patient age as well as individual VAT functional connectivity, revealed the Periaqueductal Grey Matter (PAG) to be the only structure that, together with age, significantly predicted more than half of the variance in individual symptom improvement (R=0.72, R2=0.52, p=1.86e-05, Figure 6B, Table 2), retaining significance during leave-one-out cross-validation (LOOCV) (R=0.65, R2=0.42, p=4.406e-5; Figure 6C). Additionally, we investigated if functional connectivity of the VATs with any 2 brain areas, in addition to patient age, would improve the estimation of clinical outcome (Supplementary File 1). Indeed, connectivity with PAG and limbic structures, namely the amygdala (R=0.75, R2=0.57, p=4.20e-07), anterior cingulate cortex (rostral: R=0.76, R2=0.58, p=2.75e-07; dorsal: R=0.75, R2=0.57, p=3.96e-07), bed nucleus of the stria terminallis ( R=0.76, R2=0.58, p=2.75e07), left nucleus accumbens (R=0.76, R2=0.58, p=8.51e-07), right orbitofrontal cortex (R=0.76, R2=0.58, p=2.75e-07) and right fusiform gyrus (R=0.75, R2=0.56, p=4.67e-07) was superior in predicting outcome.”

Methods, section: Statistical Analysis, pg 18, 2nd paragraph:

“Additive linear models were used to test our ability to estimate the percentage of improvement from baseline using age and the functional connectivity of all brain areas functionally connected with the VATs.”

Discussion, pg 11, 2nd paragraph:

“The functional connectivity mapping showed that the successful VATs were functionally connected to several key areas within the neurocircuitry of aggressive behaviour, including the amygdala, nucleus accumbens, hypothalamus, periaqueductal gray, and cingulate cortex. To further explore these results, we investigated the relationship between the functional connectivity of each of these brain structures and individual treatment outcomes and observed that functional connectivity with the PAG is the most relevant. In addition, when modelling the functional connectivity of any two brain areas identified in the functional connectivity analysis, we observed that the most relevant connections involved the PAG and limbic structures believed to be at the core of the neurocircuitry of aggressive behaviour, such as the amygdala, anterior cingulate cortex, and nucleus accumbens. In line with our findings, previous studies have shown positive outcomes following ablative and neuromodulation treatments targeting the amygdala, hypothalamus, and nucleus accumbens for the control of aggressive behaviour 1,3,6,67. Moreover, PAG DBS – primarily performed to treat neuropathic pain 68 – has been reported to modulate mood and anxiety 69.”

3) Functional connectivity mapping: how were R-maps generated? The authors mention that patient-specific R-maps were p-thresholded and corrected for multiple comparisons, but it is not clear how group-level maps were generated. How did the authors perform regression on these maps? Were voxels that did not survive thresholding excluded?

This is a multiple-step analysis. First, it is necessary to localize the electrodes in each patient’s brain and estimate the volume of activated tissue (VAT) with the stimulation parameters associated with a symptomatic improvement. The VATs are then used as seeds for the next steps, during which we investigate how much functional influence the VTAs have on the other areas of the brain (i.e. individual r-map). This is done by correlating the BOLD time course of the VATs seed with the BOLD time course of all other voxels in the brain. The individual r-maps are then corrected for multiple comparisons to exclude voxels with potentially spurious correlations, resulting in an individual r-map that only included voxels surviving Bonferroni correction at the level of p<0.05. Finally, to create group-level maps, a voxel-wise linear regression analysis is performed to investigate for each voxel of the map if more or less influence (corrected individual r-map with the functional connectivity of the patient’s VAT) is more or less related to the clinical outcome (i.e. individual improvement). The last step is permutation correction resulting in a significant group-level functional connectivity map (p_permute_<0.05). We improved the text in the methods section to better describe the generation of the group-level maps and added a new Figure 1—figure supplement 1 illustrating this analysis.

Methods, section: Imaging Connectomics Analyses – Structural and Functional Connectivity Mapping, pg 16, 2nd paragraph:

“For generating group-level maps, a voxel-wise linear regression analysis investigating the relationship between the functional connectivity of the VATs and the individual clinical outcome was then performed, followed by non-parametric permutation analysis. The individual improvement scores were randomly assigned to a functional connectivity map, and the analysis was repeated 1000 times. At each voxel, p_permute_<0.05 was used as a significance threshold for the final functional connectivity analysis.Figure 1—figure supplement 1 illustrates this process.”

4) The authors determined that age was a significant predictor of the outcome, but it is unclear whether certain age groups presented with distinct etiologies underlying their aggressiveness. For example, aggression in epilepsy may show a better response to DBS as opposed to schizophrenia. How does patient outcome change when stratifying according to etiology? How does model performance change when controlling for etiology? The authors should include the etiology of aggressiveness in Table 1.

This is a very interesting point. We observed a similar distribution between the pediatric and adult populations in relation to the most common etiologies. Epilepsy was the most frequent diagnosis in both populations (pediatric: 50%, adult: 62%), followed by autism spectrum disorder (pediatric: 34%, adult: 24%). The remaining etiologies were mostly composed of single cases. The percentage of intellectual disability was also similar in pediatric and adult populations. Severe deficits were observed in 75% of pediatric and 85% of adult patients, while moderate disability was present in 25% of pediatric and 15% of adult patients. The different diagnoses presented by the patient are reported in the legend of Table 1. This strategy was adopted, rather than presenting specific diagnoses in each patient’s row to preserve anonymity. In the end of Table 1, the diagnoses are reported from more to less frequent. We have added a sentence to the Results and Discussion sections as follows:

Results, section: Patients Included, pg 5, 3rd paragraph:

“The most frequent diagnoses were epilepsy (pediatric: 50%, adult: 62%) and ASD (pediatric: 34%, adult: 24%). Intellectual disability was observed in all cases. This was described as severe in 82% of patients (pediatric: 75%, adult: 85%) and moderate in 18% of patients (pediatric: 25%, adult: 15%). Table 1 presents demographic data.“

Discussion, pg 9, 3rd paragraph:

“Although we did not observe a difference in the main diagnosis between the pediatric and adult populations, the former presented the highest percentage of symptom improvement following pHyp-DBS treatment.”

5) Stimulation parameters. The authors report average pulse widths of 219 µs and 142µs respectively, which is up to 4-fold higher as compared to DBS settings used conventionally in movement disorders and will significantly alter the volume of activated tissue. Did the authors account for the drastic increases in pulse width during VAT modeling?

We thank the reviewer for raising this important point about the volume of activated tissue (VAT) modelled and the unusual pulse width observed in some patients in this cohort. These patients presented sympathetic side effects of stimulation when DBS was set with higher frequencies (e.g. increased heart rate and blood pressure). The final parameters selected were the ones that would lead to clinical benefits without generating side effects.

A multitude of models exists for estimating VATs, ranging from advanced axon cable models – the gold standard, which simulate axon membrane dynamics and require patient-specific diffusion-weighted imaging and tremendous computing power ^1^ – to simple heuristics-based models that estimate the rough extent of a VAT based on stimulation parameters without constructing an actual spatial model ^2–4^. The model employed in the current paper (and a number of previous publications by our group ^5–10^) is the FieldTripSimBio ‘E-field norm’ finite element method (FEM) model. This model, which was first described by Horn et al. ^11^ and is freely available in Lead-DBS (https://www.lead-dbs.org/), strikes a balance between the sophisticated axon cable models and the simpler heuristic models. In particular, this model constructs an electric field (E-field, by applying an electric field strength threshold, or activation threshold) and calculates a VAT associated with specific voltage settings and contact configurations, taking into account the conductivity of surrounding brain tissue and electrode components. Notably, studies comparing VAT modelling techniques ^12^ showed that ‘E-field norm’ FEM models closely approximate (<0.1 mm difference) the gold standard axon cable models in terms of the size of VATs constructed for monopolar stimulation settings. However, it should be acknowledged that the FieldTripSimBio model in Lead-DBS does not allow the user to specifically enter values for stimulation pulse width, instead employing a standard activation/electric field strength threshold (0.2 V/mm) that reflects a combination of commonly modelled axon diameters (roughly 3.5 μm) and commonly used pulse width values (i.e., 60-90 μs). This threshold is based on work by researchers such as Astrom et al. ^13^ and reflects a ‘middle ground’ value reflective of the fact that any VAT model will necessarily be an imperfect approximation of how electrical stimulation interfaces with brain tissue, depending heavily on aspects such as the diameter of local axons. Nonetheless, it is understood that increased pulse width does meaningfully increase the effective range of stimulation (thus translating to a larger VAT) by lowering the activation threshold of nearby axons ^12^.

Given that our patient cohort included a small number of patients who were stimulated with notably higher pulse widths than the pulse width value assumed by our model (90 μs), it is reasonable to consider whether we underestimated the VAT in these patients. To address this, we experimented by modelling these patients’ VATs using a simpler heuristic model ^2^ that does allow specific pulse width values to be selected by the user; specifically, we computed a range of VATs for these patients using varied pulse width values (ranging from 90 μs up to their actual pulse width values). Not surprisingly, this endeavour did yield larger VATs when higher pulse width values were used: on average, the absolute difference in VAT diameter between 90 μs and 450 μs (the largest pulse width observed in this cohort) versions of these patients’ VATs was 2 mm. To check whether this difference could have potentially impacted our results, we next repeated our probabilistic mapping analysis using altered VATs (specifically, VATs that were enlarged by 2 mm in diameter) for the patients with notably higher pulse width values. This new repeat analysis yielded a very similar average map to the original analysis: the overall map pattern and location/values of the peak corresponding to the most efficacious area for maximal symptom alleviation remaining unaltered, and only a few voxels on the periphery of the map changing in value by a couple of percentage points. As such, this new supplementary analysis indicates that our results are not meaningfully altered by the unusual pulse width observed in these patients.

We improved the Methods section of the manuscript and added a new Figure 3—figure supplement 2 illustrating both voxel efficacy maps.

Methods, section: Electrode localization and Volume of Activated Tissue (VAT) modelling, pg 15, 1st paragraph:

“Considering that increased pulse width increases the effective range of stimulation (thus translating to a larger VAT) by lowering the activation threshold of nearby axons 97, and given that our patient cohort included a small number of patients who received higher pulse widths than the ones assumed by the FieldTripSimBio model in Lead-DBS, we performed a supplementary analysis to investigate changes in VAT volumes by also modelling the pulse width observed in these patients. To this aim, we used a simpler heuristic model 38 that takes pulse width into account to compute additional VATs using pulse width values ranging from 90μs up to the patient’s actual pulse width values. As expected, this analysis yielded larger VATs when higher pulse width values were used, with an absolute difference in VAT diameter between 90 μs and 450 μs (the highest pulse width observed in this cohort) of 2 mm. Then, we investigated whether or not these larger VATs could have potentially impacted our results by performing a new probabilistic mapping analysis using the newly generated VATs (specifically, the largest VATs that were enlarged by 2 mm in diameter) for the patients with higher pulse width values. This new analysis yielded a very similar average map to the original analysis, with the overall map pattern and location/values of the peak corresponding to the most efficacious area for maximal symptom alleviation remaining unaltered with only a few voxels on the periphery of the map changing in value by a couple of percentage points (Figure 3—figure supplement 2). As such, this supplementary analysis indicates that our results were not meaningfully altered by the unusual pulse width observed in these patients.”

6 & 7. Imaging transcriptomics. The methods described lack detail: How did the authors account for differences in expression across donors, samples, and regions during preprocessing of the Allen Human Brain Atlas? How was expression data collapsed into regions of interest? Did the authors apply any normalization? Recent publications have introduced reproducible workflows for processing and preparing the AHBA expression data for analysis that is publicly available. 'genes with similar patterns of spatial distribution to the TFCE map were compiled in an extensive list'. It is unclear why authors used TFCE maps for spatial transcriptomics as opposed to the functional connectivity map featured in Figure 5. How was similarity measured between the TFCE map and the AHBA? How were candidate genes identified? Please provide a more comprehensive description of the analysis pipeline.

We would like to apologize for the short description of this analysis. We performed a gene set analysis using the abagen toolbox (https://abagen.readthedocs.io/en/stable/index.html) to investigate genes whose spatial pattern is similar to the pattern observed for clinically relevant functional connectivity. For this analysis, we used the Allen Human Brain Atlas (https://alleninstitute.org/) microarray data describing the cortical, subcortical, brainstem and cerebellar localization of over 20,000 genes in the human brain (3702 anatomical locations from 6 neurotypical adult brains) ^14–17^, along with a cell-specific aggregate gene set ^18^. These data are provided preprocessed, with gene expression values, normalized across all donors’ brains, and registered to standard MNI space, allowing for the direct comparison between the spatial pattern of gene expression and the functional connectivity map (https://human.brain-map.org/microarray/search) ^15^. The TFCE maps were used to create clusters of clinically relevant functional connectivity that have a spatial extent that overlaps with the anatomical locations where microarray data was obtained. We parcellated both datasets (results of functional connectivity analysis and Allen Gene Atlas) according to the Harvard-Oxford brain atlas and correlated the spatial distribution of gene expression with the spatial distribution of the results of the functional connectivity mapping. The resulting list of candidate genes was used as input in gene ontology tools to investigate the associated biological processes and cell types. It is important to highlight that this process involves 2 corrections for multiple comparisons using FDR at q<0.005; one correction occurs at the level of the gene list to include only the most significant genes in the gene ontology analysis; and a second correction at the level of the gene ontology analysis to consider only the most significant biological processes. We improved the Methods section as follows:

Results, section: Imaging Transcriptomics – Gene set analysis, pg 8, 3rd paragraph:

“Finally, to investigate neural phenotypes and possible neurobiological mechanisms of treatments, we performed imaging transcriptomics analysis using the abagen toolbox (https://abagen.readthedocs.io/en/stable/index.html) and the human gene expression data from the Allen Human Brain Atlas (https://alleninstitute.org/). We investigated genes with a spatial distribution of expression that resembled the pattern of brain regions with clinically relevant (following TFCE correction) functional connectivity to the stimulation locus (qFDRcor<0.0001).”

Discussion, page 12, 2nd paragraph:

“To further explore the dataset pooled in our study, we performed a spatial imaging transcriptomic analysis. Our goal was to investigate spatial correlations between gene expression patterns (i.e. microarray data from the Allen Human Brain Atlas) and imaging characteristics (i.e. functional connectivity map derived from the patient’s VATs) and possible molecular characteristics associated with symptom improvement (for a review on imaging transcriptomics see Fomito et al., 2019 70)”

Methods, section Imaging Transcriptomics – Gene Set Analysis, pg. 17, 2nd paragraph:

“We performed a gene set analysis using the abagen toolbox (https://abagen.readthedocs.io/en/stable/index.html) to investigate genes whose spatial pattern distribution is similar to the pattern of clinically relevant functional connectivity. For this analysis, we used the Allen Human Brain Atlas (https://alleninstitute.org/) microarray data describing the cortical, subcortical, brainstem and cerebellar localization of over 20,000 genes in the human brain (3702 anatomical locations from 6 neurotypical adult brains) 89–92, along with a cell-specific aggregate gene set 49. These data are provided preprocessed, with gene expression values normalized across all donors’ brains, and registered to standard MNI space allowing for the direct comparison between the spatial pattern of gene expression and the functional connectivity map (https://human.brain-map.org/microarray/search) 90. To achieve this spatial mapping, we first applied Threshold Free Cluster Enhancement (TFCE) to the functional connectivity map to generate clusters of significant functional connectivity with a spatial extent that overlaps with the anatomical locations where microarray data was obtained. Then, based on the Harvard-Oxford Atlas (http://www.cma.mgh.harvard.edu/), a regional parcellation covering 48 cortical and 21 subcortical areas in each hemisphere (a total of 138 parcellations) was applied to both datasets. A direct correlation between the spatial distribution of gene expression and functional connectivity analysis in each specific brain area was performed and corrected for multiple comparisons using FDR at qFDAcor<0.005, resulting in a gene list composed of 1734 genes associated with biological processes and ratios of genes per cell type. This list was used as input for gene ontology analyses using the EnRichr Gene Ontology tool (https://maayanlab.cloud/Enrichr/), along with the Jensen Gene Ontology enrichment tool (https://jensenlab.org/resources/proteomics/) and the Kyoto Encyclopedia of Genes and Genomes (KEGG pathway enrichment; https://www.genome.jp/kegg/kegg1b.html), resulting in an extensive list of associated biological terms that were corrected for multiple comparisons using FDR at q_FDAcor_<0.005.”

(8 & 9) What do the bar plots in Figure 7 (left) represent? P-values? The authors should label the axes to make this clear to the reader. Interprétation of imaging transcriptomics: The authors identify a therapeutic circuit associated with deep brain stimulation of the posterior hypothalamic area, however, it is unclear how to reconcile genes associated with hormones, inflammation, and plasticity in this context. The authors mention and discuss genes implicated in hormonal processing, specifically oxytocin. The results provided in Figure 7, however, do not support this finding and it is unclear how the authors identified genes linked to oxytocin. In addition, the authors identified reductions in the number of microglia and astrocytes, while oligodendrocytes were overexpressed relative to the expected distribution of genes per cell type. These findings were attributed to DBS effects, however, both connectomic and transcriptomic data are acquired from healthy subjects, which suggests a physiological deficit/enrichment in a therapeutic circuit. How do the authors interpret findings given that no electrode implantation and stimulation were performed?

The analysis of normative datasets (functional and structural connectomics and spatial transcriptomics) is based on the idea of understanding underlying mechanisms of treatment considering our current knowledge of the average human brain. Unlike patient-specific studies where imaging is acquired from a single patient or genetic profiles are extracted from tissue samples, these normative analyses rely on high-quality “atlases” derived from healthy subjects, and, in the case of the functional and structural connectivity, atlases that are calculated from very large cohorts of subjects (around 1000 brain scans). Thus, imaging connectomics investigates the pattern of brain activity and structural connectivity related to a specific area of the brain (in this case, the volume of tissue activated (VATs) with DBS) and correlates the acquired data with clinical outcomes to shed light on potential mechanisms of action. Similarly, the spatial transcriptomic analysis was used to identify spatial correlations between gene expression patterns and brain characteristics detected by MRI ^19^ (in this case, the spatial pattern of functional connectivity) and investigate possible genetic underlying mechanisms. It is important to highlight that previous studies have shown that normative analyses yield results that are similar to the ones observed in patient-specific data ^20–22^. In the case of imaging connectomics, normative datasets have been used to create probabilistic models of optimal connectivity associated with patients’ outcomes that are meaningful to predict outcomes in patient-specific connectivity data ^21^. Thus, these exploratory data-driven approaches strive to simulate the presumed fingerprint that a particular patient’s individualized DBS intervention might modulate and allows for the investigation of possible mechanisms of action in a large, previously inaccessible cohort of patients whose individual data are available. We improved the Discussion section of the manuscript. We also would like to apologize for the previous Figure 7, which was incorrect. We included the label for the bar plots in the left panel to facilitate the reading of the graph and added the missing result from the KEGG 2021 Human Library that shows the oxytocin signalling pathway.

Discussion, pg 10, 2nd paragraph:

“The analysis of functional and structural connectivity via normative datasets is an opportunity to investigate possible underlying mechanisms based on the current knowledge of the typical human brain. Unlike patient-specific studies, where functional MRI and diffusion-weighted images are acquired from a single patient, normative analyses rely on high-quality atlases derived from very large cohorts of healthy subjects (around 1000 brain scans). Thus, these exploratory data-driven approaches are intended to investigate the pattern of brain activity and structural connectivity related to a specific region of interest (i.e. individual VAT) and correlate these data with clinical outcomes to simulate the presumed brain changes associated with DBS treatment. This approach allows for the expansion of the investigation to a large, previously inaccessible cohort of patients whose individual data are available, generating knowledge that can help optimize treatment in future patient populations. Although normative connectivity data may not fully capture patient- or pathology-specific variations, they have been previously shown to generate results that are comparable to patient-specific imaging data 62,63.”

Discussion, pg 11, 2nd paragraph:

“The functional connectivity mapping showed that the successful VATs were functionally connected to several key areas within the neurocircuitry of aggressive behaviour, including the amygdala, nucleus accumbens, hypothalamus, periaqueductal gray, and cingulate cortex. To further explore these results, we investigated the relationship between the functional connectivity of each of these brain structures and individual treatment outcomes and observed that functional connectivity with the PAG is the most relevant. In addition, when modelling the functional connectivity of any two brain areas identified in the functional connectivity analysis, we observed that the most relevant connections involved the PAG and limbic structures believed to be at the core of the neurocircuitry of aggressive behaviour, such as the amygdala, anterior cingulate cortex, and nucleus accumbens. In line with our findings, previous studies have shown positive outcomes following ablative and neuromodulation treatments targeting the amygdala, hypothalamus, and nucleus accumbens for the control of aggressive behaviour 1,3,6,67. Moreover, PAG DBS – primarily performed to treat neuropathic pain 68 – has been reported to modulate mood and anxiety 69.

Discussion, pg 12, 3rd paragraph:

“A large number of genes identified by the Jensen Compartments and KEGG 2021 Human Libraries are part of biological processes associated with neuronal communication and plasticity, thus possibly linked to the intrinsic mechanism of action of DBS 76,77. As a neuromodulation therapy, DBS induces long-lasting changes in cellular and molecular aspects of neurons belonging to dysfunctional neurocircuitries in order to restore functional normality 12,78,79. Although its neurobiological mechanisms are not fully understood, DBS is believed to exert its effects by altering the cellular membrane potential, resulting in either an increase or a decrease in action potentials 12,78,79. These changes are then propagated throughout neural networks, changing neurotransmitter dynamics, protein expression, and membrane receptor availability 12,78,79. In fact, a previous study using imaging transcriptomics to investigate brain changes after functional neurosurgery of the amygdala in patients with refractory aggressive behaviour has also identified several genes related to neuronal communication and plasticity 1. Preclinical studies have also found significant increases in neural precursor cells, plasticity, and precursor cell markers in animals implanted with DBS, along with significant reductions in the number of activated microglia and astrocytes 80–84. This is in line with the cell type investigation, where a significant reduction in the ratio of genes per cell type was observed for astrocytes and microglia when compared to the expected distribution. Both astrocytes and microglia are highly implicated in mechanisms of neuroinflammation 85, which in turn is associated with several psychiatric disorders 86. The reduction of neuroinflammatory markers (such as glial activation and interleukin levels) has been described in several preclinical DBS studies and is thought to be involved in the mechanism of action of DBS 82,87,88.”

10) Data availability. Code used for data processing should be made openly available or shared as source data along with the Figures that were generated using the code. Sweet-spot, structural, and functional connectivity maps should be shared openly.

All tools and codes necessary for localizing the electrodes, the estimation of the volume of activated tissues, and imaging connectomics analyses are freely available in Lead-DBS (https://www.lead-dbs.org/), a toolbox designed for DBS electrode reconstructions and computer simulations based on postoperative imaging. All codes for spatial transcriptomics are freely available in abagen (https://abagen.readthedocs.io/en/stable/), a toolbox designed for analyzing the Allen Brain Atlas genetics data. Along with the codes, the websites for these tools provide manuals describing the step-by-step procedure for successful analysis. The datasets are freely available at Zenodo (doi: 10.5281/zenodo.7344268). We improved our Data Availability Statement to address this concern.

Data availability statement, pg 18, 4th paragraph:

“The codes for electrode localization, modelling of the volume of activated tissue, and imaging connectomics (i.e. functional and structural connectivity) are freely available in Lead-DBS (https://www.lead-dbs.org/). The codes, along with the Allen Human Brain Atlas (AHBA) microarray dataset, for the analysis of spatial transcriptomics are freely available in abagen (https://abagen.readthedocs.io/en/stable/). Along with the codes, the websites for these two toolboxes provide manuals describing the step-by-step procedure for successful analysis. The dataset accompanying this study is freely available at Zenodo (doi: 10.5281/zenodo.7344268).”

11) Methods: Given the small number of patients over 5 centers, the authors should better address the similarities and differences of data collection across the sites.

We added the similarities and differences across sites in the first paragraph of the Results section, as follows:

Results, section Patients Included, pg 6, 1st paragraph:

“In all centres, patients were evaluated by a multidisciplinary healthcare team that reviewed all clinical and medication history. Patients were only considered for surgery when a consensus was reached that they indeed presented severe medication-resistant aggressive behaviour (i.e., persistent severe symptomatology despite using multiple medications at well-established doses and duration). Whole-brain T1-weighted MRI was acquired preoperatively for surgical planning (1.5T MRI used in 14 cases at 3 centres, and 3T MRI used in 19 cases at 2 centres). Postoperative brain MRI and/or CT were obtained for electrode localization. Aggressive behaviour was assessed using standard questionnaires conducted by a neuropsychologist and answered by the parents/caregivers in all centres (i.e., Overt Aggressive Behaviour [OAS] used in 3 centres; Modified Overt Aggressive Behaviour [MOAS] used in 1 centre; Inventory for Client and Agency Planning [ICAP] used in 1 centre). Treatment response is reported as the percentage of improvement at the last follow-up relative to baseline (preoperative). Patients presenting more than 30% improvement were considered to be treatment responders.”

12) Results: Clinically relevant bundles included somatosensory, emotions, and motor connections. This is pretty much a summary of connections through the hypothalamus and surrounding regions. Do the authors suggest that the effectiveness of DBS in this region is this nonspecific?

This is a very interesting point. As the VATs are located in the posterior hypothalamus, the only fibres that could be touched by the VATs are the ones that cross the posterior hypothalamus. However, not all the fibres that cross the posterior hypothalamus are relevant for symptom improvement, as our statistical analysis only includes the most relevant fibre tracts. Understanding how symptom improvement and the location of the most efficacious stimulation region correlate with these tracts is important to refine surgical planning, as one could use individual patient tractography before surgery to determine the optimal location of the electrode based on individual fibre tracts. It is worth mentioning, however, that this analysis is excellent for segregating the fibre tracts as relevant or not relevant, but it is not capable of tearing apart the system to determine which of those are necessary for symptom alleviation. This could be done in preclinical settings using opto-or-chemogenetics to selectively inactivate specific fibre tracts and investigate the impact on behaviour and network dynamics. Thus, the structural connectivity analysis presented here contributes to the body of knowledge on the network of aggressive behaviour and provides clinically relevant data that can be useful to improve future patient outcomes. We added a sentence to the Discussion-limitations section of the manuscript addressing this concern, as follows:

Discussion, pg 13, 3rd paragraph:

“The structural connectivity analysis performed here provides a detailed overview of the relevant tracts that are related to symptom alleviation in this patient population. However, for precise segregation of these fibre tracts and the determination of those that are necessary for symptom improvement, it would be necessary to perform preclinical experiments to selectively activate/inactivate specific fibre tracts and investigate the impact of these changes on behaviour and network dynamics. Nevertheless, understanding how the location of the most efficacious area of stimulation correlates with these tracts is important for refining the surgical planning of future patients, as one could use individual patient tractography to determine the optimal location of the electrodes based on how it would impact individual fibre tracts.”

13) Indeed, it appears that the motor system is most involved based on the tractography. This is consistent with figure 3 in which the 'sweet spot' appears to be lateral to the hypothalamus and more centered in the red nucleus/STN region (Figures 1 and 3).

Although the structural connectivity mapping revealed tracts involved in motor and sensory information, it also showed tracts known to be involved in the regulation of emotions, such as the MFB, the Amygdalofugal Pathway and the ALIC. As discussed above, it is not possible to determine that motor projections are stronger or more relevant than others as we do not have the means to segregate the influence of each of these tracts in symptom improvement. However, we agree with the reviewer that the engagement of the motor system is indeed highly relevant for the reduction of aggressive behaviours, as we have previously shown that aggressive behaviour is highly correlated with motor agitation ^23,24^. Additionally, in the context of ASD, self-injury behaviour is defined as a type of repetitive/stereotypic behaviour that results in physical injury to the patient’s own body. We added a sentence to the Discussion section highlighting the relevance of the motor system. The sweet spot is indeed located at the posterior-inferior-lateral region of the posterior hypothalamic area, reaching the most superior part of the red nucleus, however, not including the STN. It is important to highlight that the voxel-efficacy mapping only shows voxels that are associated with a minimum of 50% symptom improvement following treatment, thus, the areas not touching the red nucleus are also associated with excellent symptom alleviation. We added a new Figure 3—figure supplement 1, to show the localization of these areas in relation to the STN and red nucleus.

Discussion, pg 10, 3rd paragraph:

“Structural connectivity analysis in our study implicated fibre tracts involved in somatosensation (e.g. medial lemniscus and spinothalamic tract), emotional regulation (e.g., amygdalofugal pathway and medial forebrain bundle), and motor signalling (e.g., motor projections and central tegmental tract) in the amelioration of aggressive behaviour. Previous studies have shown a significant correlation between aggressive behaviour and motor agitation 5,64, with a concomitant reduction in both following neuromodulatory treatments targeting the posterior hypothalamus 7,37. Indeed, patients with neurodevelopmental disorders, especially those presenting with self-injurious behaviour, are known to frequently have altered sensory perception (including reduced pain sensitivity), which can contribute to the chronicity and severity of the behaviour 1,65. Furthermore, self-injurious behaviours in this patient population are considered to be a type of repetitive/stereotypic behaviour that results in physical injury to the patient’s own body 66,67, highlighting the relevance of the motor system in the network subserving aggressive behaviour. As such, modulation of somatosensory and motor pathways – perhaps leading to increased pain perception or awareness, and reduction in repetitive/stereotypic behaviours – could help reduce self-injurious behaviours. ”

Discussion section, pg. 10, 1st paragraph:

“One of the most commonly targeted areas in those series lies along the midpoint of the line between the anterior and posterior commissures (AC-PC line), in a region anterior to the rostral end of the aqueduct, posterior to the anterior border of the mammillary body and superior to the red nucleus 3,34 (Figure 3—figure supplement 1 shows the location of the sweet-spot in relation to the red nucleus and subthalamic nucleus)”

(14 & 15) The emphasis on connections on monoamine connections (P. 7) is unwarranted, given the stronger connections to motor regions. Moreover, both DBS and tractography favor myelinated fibers over unmyelinated ones. The monoamines are unmyelinated. Finally, the MFB is a relatively small bundle compared to others in the region that carry motor and sensory information to and from the brainstem. The resolution of tractography is unlikely to distinguish the MFB and favor other fiber tracts. In this regard, the paragraph on P. 9, "functional connectivity…" is confusing. (My understanding is that the authors are linking connections to the amygdala, etc.) via the monoamines. Is that correct? If so, that's a discussion point and not well supported by the connectivity data.

We would like to apologize for not being clear in the discussion of our findings. Although functional and structural connectivity maps are related, they provide different means of exploring distinct aspects of the connectivity profile of each VAT. While the structural connectivity map may help to explain symptom improvement via direct fibre modulation (i.e. fibres that touch vs fibres that do not touch the VAT), the functional connectivity map investigates the functional dynamics of the network via BOLD signals (functional MRI). In this manuscript, we showed the functional connectivity (not fibre tracts) of the VATs with areas known to regulate monoamine production, such as the Raphe nuclei and the Substantia Nigra. Both serotonin and dopamine are critically involved in the control of aggressive behaviours being the target of the main classes of medications used to treat aggressive behaviours in different patient populations. We improved our Discussion section to address this concern as follows:

Discussion, pg 11, 2nd paragraph:

“The functional connectivity mapping showed that the successful VATs were functionally connected to several key areas within the neurocircuitry of aggressive behaviour, including the amygdala, nucleus accumbens, hypothalamus, periaqueductal gray, and cingulate cortex. To further explore these results, we investigated the relationship between the functional connectivity of each of these brain structures and individual treatment outcomes and observed that functional connectivity with the PAG is the most relevant. In addition, when modelling the functional connectivity of any two brain areas identified in the functional connectivity analysis, we observed that the most relevant connections involved the PAG and limbic structures believed to be at the core of the neurocircuitry of aggressive behaviour, such as the amygdala, anterior cingulate cortex, and nucleus accumbens. In line with our findings, previous studies have shown positive outcomes following ablative and neuromodulation treatments targeting the amygdala, hypothalamus, and nucleus accumbens for the control of aggressive behaviour 1,3,6,67. Moreover, PAG DBS – primarily performed to treat neuropathic pain 68 – has been reported to modulate mood and anxiety 69.

Another aspect detected by the functional connectivity analysis was the involvement of areas responsible for the production of serotonin (i.e., raphe nuclei) and dopamine (i.e., substantia nigra). Both these monoamines are thought to be directly involved in the initiation and maintenance of aggressive behaviours due to their role in facilitating top-down inhibitory controls, regulating mood, and mediating social behaviours 3,7,70. Antipsychotics (both typical and atypical) and selective serotonin reuptake inhibitors – the most commonly used drugs to control of aggressive behaviours – act on the dopaminergic and serotonergic neurotransmission systems 1,3. In a mouse model of escalated aggression, pHyp-DBS increased the density of serotonin receptors type 1A (5-HT1A) in the orbitofrontal cortex and amygdala, while pretreatment with a 5-HT1A antagonist countered the anti-aggressive effect of stimulation 71.”

16 & 17) Other comments: The introduction lacks clear text referring to the organization and connections of the hypothalamus based on classic literature. Examples. First, the authors described the division into 3 regions: anterior, medial, and posterior (reference is the author's article in neurosurgery. Then, without reference to these three regions, describe preclinical studies of the lateral and ventromedial hypothalamus. How do these two areas relate to the three regions?

We improved the Introduction section to better describe the anterior-posterior subdivision of the hypothalamus, as follows:

Introduction, pg 4, 2nd paragraph:

“The hypothalamus is a diencephalic structure with well-established roles in the control of homeostasis and motivated behaviours and is a key area in a broader neurocircuitry regulating aggressive behaviour that also involves the orbitofrontal cortex, hippocampus, amygdala and periaqueductal gray 24–26. Along the anterior-posterior axis, the hypothalamus can be divided into three regions (i.e. supraoptic or anterior, tuber cinereum or medial and supramamillary or posterior) with distinct cell types, projections and functions 24,27. While the anterior region is mainly involved in thermoregulation and the control of circadian rhythms, the medial region regulates feeding and sexual behaviour and plays a critical role in several endocrine and autonomic processes 24,28,29. The posterior hypothalamus (pHyp) is an ergotropic area involved in the generation of sympathetic responses 30. It provides robust projections to the midbrain and reticular formation via the hypothalamotegmental tract, thus being critical in the regulation of wakefulness and stress responses 27,31,32. Indeed, lesions (e.g., hamartomas and gliomas) located at the pHyp have been reported to cause apathetic and somnolent symptoms, while the selective neurosurgical ablation of a small portion of the pHyp has been successfully used to treat patients with severe treatment-resistant aggressive behaviour (for a detailed review, see Gouveia et al. 2019 3,33,34). More recently, DBS of the pHyp (pHyp-DBS) has also been shown to reduce aggressive symptoms in humans, with variable outcomes 7–10,35–37.”

References:

1. Gunalan, K., Howell, B. & McIntyre, C. C. Quantifying axonal responses in patient-specific models of subthalamic deep brain stimulation. Neuroimage 172, 263–277 (2018).

2. Dembek, T. A. et al. Probabilistic mapping of deep brain stimulation effects in essential tremor. Neuroimage Clin 13, 164–173 (2017).

3. Kuncel, A. M., Cooper, S. E. & Grill, W. M. A method to estimate the spatial extent of activation in thalamic deep brain stimulation. Clin. Neurophysiol. 119, 2148–2158 (2008).

4. Mädler, B. & Coenen, V. A. Explaining clinical effects of deep brain stimulation through simplified target-specific modeling of the volume of activated tissue. AJNR Am. J. Neuroradiol. 33, 1072–1080 (2012).

5. Elias, G. J. B. et al. Probabilistic Mapping of Deep Brain Stimulation: Insights from 15 Years of Therapy. Ann. Neurol. 89, 426–443 (2021).

6. Germann, J. et al. Brain structures and networks responsible for stimulation-induced memory flashbacks during forniceal deep brain stimulation for Alzheimer’s disease. Alzheimers. Dement. 17, 777–787 (2021).

7. Elias, G. J. B. et al. Mapping the network underpinnings of central poststroke pain and analgesic neuromodulation. Pain 161, 2805–2819 (2020).

8. Gouveia, F. V. et al. Case report: 5 Years follow-up on posterior hypothalamus deep brain stimulation for intractable aggressive behaviour associated with drug-resistant epilepsy. Brain Stimul. 14, 1201–1204 (2021).

9. Coblentz, A. et al. Mapping efficacious deep brain stimulation for pediatric dystonia. J. Neurosurg. Pediatr. 27, 346–356 (2021).

10. M Oliveira, L. et al. Probabilistic characterisation of deep brain stimulation in patients with tardive syndromes. J. Neurol. Neurosurg. Psychiatry (2021) doi:10.1136/jnnp-2020-324270.

11. Horn, A. et al. Connectivity Predicts deep brain stimulation outcome in Parkinson disease. Ann. Neurol. 82, 67–78 (2017).

12. Duffley, G., Anderson, D. N., Vorwerk, J., Dorval, A. D. & Butson, C. R. Evaluation of methodologies for computing the deep brain stimulation volume of tissue activated. J. Neural Eng. 16, 066024 (2019).

13. Astrom, M., Diczfalusy, E., Martens, H. & Wardell, K. Relationship between neural activation and electric field distribution during deep brain stimulation. IEEE Trans. Biomed. Eng. 62, 664–672 (2015).

14. Hawrylycz, M. J. et al. An anatomically comprehensive atlas of the adult human brain transcriptome. Nature 489, 391–399 (2012).

15. Arnatkeviciute, A., Fulcher, B. D. & Fornito, A. A practical guide to linking brain-wide gene expression and neuroimaging data. Neuroimage 189, 353–367 (2019).

16. Arnatkeviciute, A., Markello, R. D., Fulcher, B. D., Misic, B. & Fornito, A. Toward Best Practices for Imaging Transcriptomics of the Human Brain. Biol. Psychiatry 93, 391–404 (2023).

17. Markello, R. D. et al. Standardizing workflows in imaging transcriptomics with the abagen toolbox. e*Life* 10, (2021).

18. Seidlitz, J. et al. Transcriptomic and cellular decoding of regional brain vulnerability to neurogenetic disorders. Nat. Commun. 11, 3358 (2020).

19. Fornito, A., Arnatkevičiūtė, A. & Fulcher, B. D. Bridging the Gap between Connectome and Transcriptome. Trends Cogn. Sci. 23, 34–50 (2019).

20. Arnatkeviciute, A., Fulcher, B. D., Bellgrove, M. A. & Fornito, A. Imaging Transcriptomics of Brain Disorders. Biol Psychiatry Glob Open Sci 2, 319–331 (2022).

21. Wang, Q. et al. Normative vs. patient-specific brain connectivity in deep brain stimulation. Neuroimage 224, 117307 (2021).

22. Elias, G. J. B. et al. Normative connectomes and their use in DBS. in Connectomic Deep Brain Stimulation 245–274 (Elsevier, 2022).

23. Gouveia, F. V. et al. Bilateral Amygdala Radio-Frequency Ablation for Refractory Aggressive Behavior Alters Local Cortical Thickness to a Pattern Found in Non-refractory Patients. Front. Hum. Neurosci. 15, 653631 (2021).

24. Venetucci Gouveia, F. et al. Case report: 5 Years follow-up on posterior hypothalamus deep brain stimulation for intractable aggressive behaviour associated with drug-resistant epilepsy. Brain Stimul. (2021) doi:10.1016/j.brs.2021.07.062.

25. Gouveia, F. V. et al. Amygdala and Hypothalamus: Historical Overview With Focus on Aggression. Neurosurgery 85, 11–30 (2019).

26. Gouveia, F. V. et al. Longitudinal Changes After Amygdala Surgery for Intractable Aggressive Behavior: Clinical, Imaging Genetics, and Deformation-Based Morphometry Study-A Case Series. Neurosurgery 88, E158–E169 (2021).

27. Yan, H. et al. Deep brain stimulation for extreme behaviors associated with autism spectrum disorder converges on a common pathway: a systematic review and connectomic analysis. J. Neurosurg. 1–10 (2022).

28. Knotkova, H. et al. Neuromodulation for chronic pain. Lancet 397, 2111–2124 (2021).

29. Gray, A. M. et al. Deep brain stimulation as a treatment for neuropathic pain: a longitudinal study addressing neuropsychological outcomes. J. Pain 15, 283–292 (2014).

30. Miczek, K. A., Fish, E. W., De Bold, J. F. & De Almeida, R. M. M. Social and neural determinants of aggressive behavior: pharmacotherapeutic targets at serotonin, dopamine and γ-aminobutyric acid systems. Psychopharmacology 163, 434–458 (2002).

31. Gouveia, F. V. et al. Reduction of aggressive behaviour following hypothalamic deep brain stimulation: involvement of 5-HT1Aand testosterone. bioRxiv (2023) doi:10.1101/2023.03.20.533520.